# Generation of autogenic knickpoints in laboratory landscape experiments evolving under constant forcing.

**Léopold de Lavaissière[1], Stéphane Bonnet[1], Anne Guyez[1], and Philippe Davy[2]**

*[1] GET, Université de Toulouse, CNRS, IRD, UPS(Toulouse), France,*

*[2] Univ Rennes, CNRS, Géosciences Rennes - UMR 6118, 35000 Rennes, France,*

Correspondence to: Stéphane Bonnet (stephane.bonnet@get.omp.eu)

**ABSTRACT**

**The upstream propagation of knickpoints in river longitudinal profiles is commonly assumed to be related to discrete changes in tectonics, climate or base-level. However, the recognition that some knickpoints may form autogenically, independent of any external perturbation, may challenge these assumptions. We investigate here the genesis and dynamics of such autogenic knickpoints in laboratory experiments at the drainage basin scale, where landscapes evolved in response to constant rates of base-level fall and precipitation. Despite these constant forcings, we observe that knickpoints regularly initiate in rivers at the catchments' outlet throughout the duration of experiments. The upstream knickpoint propagation rate does not decrease monotonically in relationship with the decrease of drainage area, as predicted by stream-power based models, instead the propagation rate first increases until the mid-part of catchments before decreasing. To investigate the dynamics of the knickpoints, we calculated hydraulic information (water depth, river width, discharge and shear stress) using a hydrodynamic model. We show that knickpoint initiation at the outlet coincides with a fairly abrupt river narrowing entailing an increase in their shear stress. Then, once knickpoints have propagated upward, rivers widen causing a decrease in shear stress and incision rate, and making the river incision less than the base-level fall rate. This creates an unstable situation which drives the formation of a new**

**knickpoint. The experiments suggest a new autocyclic model of knickpoint generation controlled by river width dynamics independent of variations in climate or tectonics. This questions an interpretation of landscape records focusing only on climate and tectonic changes without considering autogenic processes.**

**1 Introduction**

Knickpoints are discrete zones of steepened bed gradient that are commonly observed in river longitudinal profiles. Although they occasionally occur due to changes in bedrock properties (e.g. Duvall et al., 2004), in many cases they are dynamic features that propagate upstream along drainage networks (Whipple and Tucker, 1999; Kirby and Whipple, 2012; Whittaker and Boulton, 2012). In this latter case, they are commonly considered as formed in response to variations in external forcing such as uplift rate, sea level or climate (e.g. Crosby and Whipple 2006; Berlin and Anderson, 2007; Kirby and Whipple, 2012; Whittaker and Boulton, 2012; Mitchell and Yanites, 2019) which opens the possibility of using knickpoints in landscapes to identify such changes. Several studies pointed out, however, that some knickpoints could be autogenic, that is to say internally-generated without any variation in boundary conditions (e.g. Hasbargen and Paola, 2000, 2003; Finnegan and Dietrich, 2011). Understanding how knickpoints can form autogenically is therefore crucial for interpreting changes in external forcing from knickpoint occurrence in landscapes. Most observations of autogenic knickpoints formation come from experimental modelling (see for example Paola et al., 2009) their initiation being attributed to amplification of local instabilities in flume (Scheingross et al., 2019) and drainage basin scale (Hasbargen and Paola, 2000), experiments. In these latter experiments for example, successive knickpoints initiated despite constant external forcing (base-level fall and precipitation) throughout the duration of the runs, even when landscapes were at steady-state on average in terms of sediment flux. Internal processes may also complexify the propagation of knickpoints as shown in the flume experiments of Cantelli and Muto (2014) and Grimaud et al. (2016) where a single discrete event of base-level drop resulted in the propagation of multiple waves of knickpoints.

.

In this work, we consider the generation and dynamics of autogenic knickpoints in laboratory-scale
drainage basins experiments forced by constant rate of base-level fall and steady precipitation. Such
landscape experiments have been used successfully to explore how tectonics and climate impact erosion
processes and the evolution of topography under controlled conditions (e.g. Hasbargen and Paola, 2000;
Bonnet and Crave, 2003; Lague et al., 2003; Turowski et al., 2006; Bonnet, 2009; Singh et al., 2015;
Sweeney et al., 2015; Moussirou and Bonnet, 2018). This approach allows for the observation of
complex dynamics that are sometimes difficult to simulate numerically and sheds new light on the way
natural landforms may evolve. Landscape experiments capture the tree-like structure of drainage
networks, the supply of eroded material from hillslopes, and especially their fluctuations, which is a
natural complexity that is not reproduced in flume experiments, for example. The experiments presented
here have been performed using a new setup specifically designed to investigate the evolution of a large,
meter-long, single drainage basin under controlled forcing condition. In previous similar catchment-
scale experiments (Hasbargen and Paola, 2000, 2003; Bigi et al., 2006; Rohais et al., 2012) the outlet
location was pinned to a narrow motor-controlled gate used to simulate base-level fall and which also
set the river width at the outlet. A specificity of our setup here is to use a large gate instead of a narrow
one, allowing experimental rivers to freely evolve downstream, with no constraints on their width. We
report here results from experiments where successive knickpoints initiate near the outlet autogenically
and propagate within drainage basins. The experiments show a new model of autogenic knickpoint
initiation and propagation driven by downstream river width dynamics.

**2 Methods**
We present here results from 3 experiments, BL05, BL10 and BL15, performed with different rates of
base level fall, of respectively 5, 10 and 15 mm h$^{-1}$ (Table 1). The facility is a box with dimensions 100
x 55 cm filled with silica paste (Fig. 1; see also Fig. S1 in the Supplemental Material). At its front side,
a sliding gate, 41 cm-wide, drops down at constant rate, acting as the base level. The initial surface
consists on a plane with a counterslope of ~3°, opposite to the base level-side (Fig. 1C). During a run,
runoff-induced erosion occurs in response to steady base level fall and rainfall (mean rainfall rate is 95

mm h$^{-1}$ with a spatial coefficient of variation (standard deviation/mean) of 35%). Incision initiates at some point along the base level and propagates upstream until complete dissection of the initial surface. Note that the counterslope of the initial surface allows separating the rainfall flux between the base level and the opposite side of the device, creating a water divide (Fig. 1B).

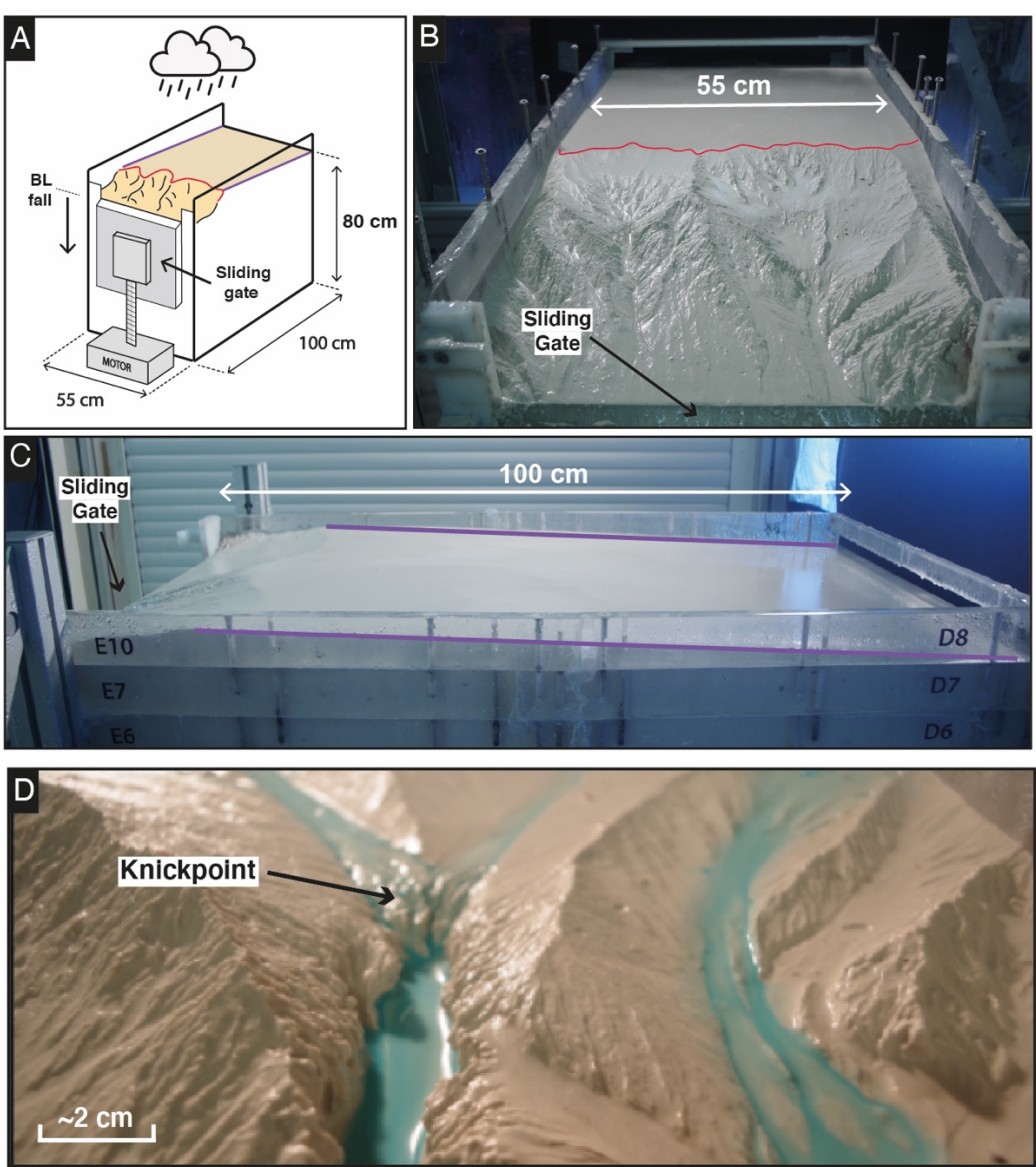

***Figure 1****. Experimental setup. Purple and red lines show respectively the counter-slope of the initial topography and the main water divide. (A) Sketch of the erosion box with the sliding gate, 41 cm wide, used to drop down the base level (BL). (B), (C) Front and side photographs (experiments BL10 at 525 min and BL15 at 185 min). (D) Photograph of a typical knickpoint studied here.*

*Table 1. Parameters of experiments*

| Experiments | Base Level Fall (mm/h) | Precipitation Rate (mm/h) | Duration Time (min) | Mean Divide Retreat Rate (mm/h) | nDDVmax* | Mean Knickpoint Retreat Rate (mm/h) |
|---|---|---|---|---|---|---|
| BL15 | 15 | 95 | 1065 | 66.3 | 0.52 | 183.6 ± 93.8 |
| BL10 | 10 | 95 | 1200 | 55.7 | 0.57 | 164.8 ± 74.8 |
| BL05 | 5 | 95 | 1455 | 25 | 0.54 | 73.1 ± 50 |

*nDDVmax : normalized distance of maximum knickpoint velocity
Experiments were stopped every 5 min to digitize the topography using a laser sheet and to construct
Digital Elevation Models (DEMs) with a pixel size of 1 mm$^2$. Longitudinal profiles and knickpoints
were extracted with a semi-automatic procedure that had to be developed to process the ~200 DEMs per
experiment. For this purpose, we first extracted longitudinal profiles by finding the lowest elevation on
successive rows (lines oriented parallel to the sliding gate) of each DEM within a 20 cm-wide swath
perpendicular to the sliding gate that included the main river (the one with the largest catchment for each
experiment). Then the lowest elevation found in our search was plotted against distance down the long
axis of the box. This procedure has already been applied by Baynes et al. (2018) and Tofelde et al.
(2019). It may result in a slight overestimation in channel slope because it does not consider the obliquity
of channels within the box in the distance calculation nor their sinuosity. However, these effects are of
minor influence here, because most channels are straight and roughly parallel to the long side of the box.
In a second step, we computed the erosion rates by considering elevation difference between each
successive pairs of longitudinal profiles and we identified knickpoints as peaks in erosion rates with
values above the steady erosion amount defined by the rate of base-level fall (Fig. 2). We verified
manually that this procedure defines knickpoints correctly by checking the computed positions on
longitudinal profiles. We investigated in particular if the procedure is robust with respect to the time
interval between successive profiles. We found that the record interval of 5 minutes is too small to
produce well-defined erosional peaks, which lead us to identify knickpoint positions from a time-interval
of 10 minutes. Then, we built a first catalogue of knickpoints positions at different times from which we
manually extract the successive positions of each individual knickpoint. We complemented the database
by computing incremental retreat rates of knickpoints from their successive positions.

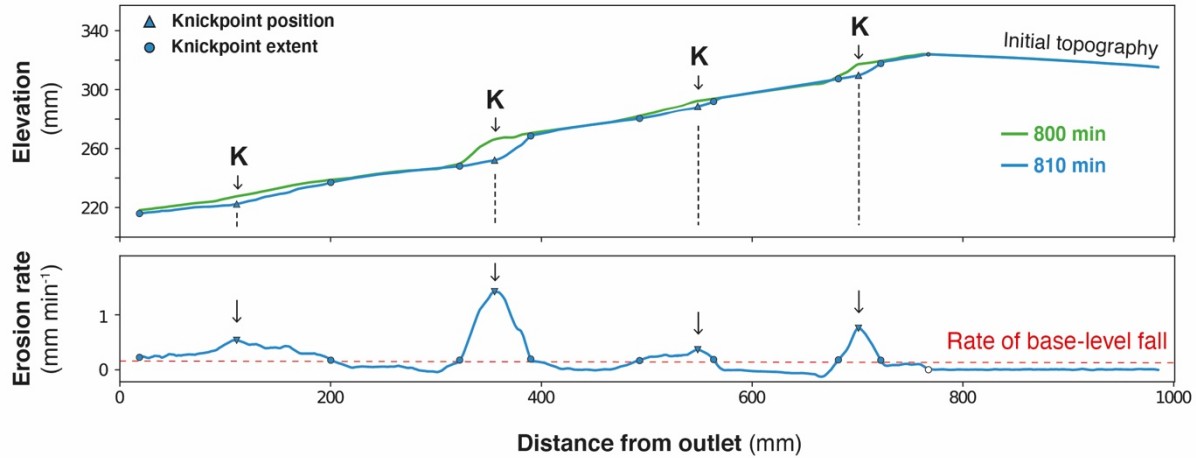


***Figure 2****. Graph showing two successive longitudinal profiles of experiment BL10 taken at 10 min*
*interval (top) and corresponding erosion rate profile (bottom). Triangles illustrate the position of*
*erosional peaks taken as knickpoint position (black arrows). Red dashed line shows the rate of base-*
*level fall.*

DEMs were also used to compute hydraulic information (water depth, river width, discharge and shear
stress) using the Floodos hydrodynamic model of Davy et al. (2017; see also Baynes et al. (2018,
2020) for previous use of Floodos for analyzing laboratory experiments). Floodos is a precipiton-based
model that calculates the 2D shallow water equations (SWE) without inertia terms, from the routing of
elementary water volumes on top of topography. We ran Floodos on successive DEMs of experiments
by inputting spatial distribution of precipitation, then generating several output raster products at the
pixel size, including water depth, unit discharge and bed shear stress that were then used for
computation of hydrologic parameters (river width, specific discharge and shear stress). The solution
of the SWE depends on the friction coefficient (C) that depends on water viscosity only for laminar
flow; its theoretical value is ~2.5 x $10^6$ $m^{-1}$ $s^{-1}$ at 10°C (Baynes et al., 2018). To ensure that Floodos
outputs (e.g. water depth raster maps) calculated using this value are consistent with actual experiment
hydraulic conditions, we injected dye in the rainfall water during a run to catch the actual extent of
water flow and make rivers visible. A visual comparison with Floodos results shows a good match
between model outputs and experimental results (Fig. S2), which validates the numerical method and
the expected theoretical friction coefficient C (Baynes et al., 2018). Given the difficulty to measure the
mm-scale water depth without perturbating the flow, river widths were extracted from Floodos DEM
outputs by thresholding the water depth mapsconsidering that river banks correspond to sharp
variations in water depth. The water depth threshold was estimated by trial and error by comparing the
the rivers extracted from the calculation with direct observations on experiments where rainwater was
colored by red dye (Fig. 3). A good visual agreement was obtained for a threshold value of the water
depth between 0.1 and 0.5 mm, and a mid-value of 0.3 mm was then used for determining river
widths.

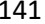


*Figure 3*. *Impact of water depth threshold used to delineate river boundaries on estimated river widths. A. Map views of water depths (blue colors) superimposed to DEM, for water depth threshold values between 0.025 and 1.5 mm. Red and purple lines show corresponding river widths for two rivers. Photo on the bottom right shows the active river width during the corresponding experimental run ("control run"), viewed by injecting red dye in the water used to generate the artificial rainfall. B. Corresponding local river widths for the two sections shown by red and purple lines. A threshold value of between 0.1 and 0.5 mm shows a good similarity between rivers on water depth map and the control run. Here, a mid-value of 0.3 mm has been chosen for computing river widths.*

## 3 Results

### 3.1 Dynamics of knickpoints retreat

In each experiment, base level fall induces the growth of drainage networks by headward erosion and the progressive migration of a main water divide (Fig. 4). The migration rate of the divide is constant in each experiment (Fig. 5 and Table 1), and this value increases from 25 to 66 mm h$^{-1}$ with prescribed rate of base level fall of 5 to 15 mm h$^{-1}$. The successive longitudinal profiles of the main river investigated in each experiment (Fig. 6) illustrate the growth of rivers as they propagate within the box. These profiles show alternations of segments with low and high slopes, the latter defining knickpoints. Knickpoints regularly initiate at the outlet throughout the duration of the runs in all experiments and propagate upward until they reach and merge with the divide, some profiles showing even several knickpoints that retreat simultaneously (Fig. 6). A characteristic of these knickpoints highlighted in Figure 7 (see also Fig. 6) is that they generally initiate downstream with a gentle slope and gradually steepen as they migrate upstream. Their maximum slope is generally reached when they have propagated to the central part of the profiles (see below). Then the slope is maintained or slightly decreases during their retreat in the upper segment of the profiles.

The mean retreat velocity of knickpoints varies between experiments from $73 \pm 50$ to $183 \pm 94$ mm h$^{-1}$ (Table 1) and increases as a function of the rate of base-level fall. Data suggest a non-linear relationship

between base-level fall rate and mean retreat velocity of knickpoints, however complementary
experiments would be necessary to constraint this dependency. To investigate the propagation of the
knickpoints, we built space-time diagrams (Fig. 8) by plotting the successive alongstream position of
each knickpoint over experimental runtime, as well as the position of the water divide in the box as
already reported in Figure 5. To compare the dynamics of knickpoints within an experiment regardless
of the stage of water divide retreat into the box, the position of knickpoints (distance to outlet, D) has
been normalized to the position of the divide, hereafter referred to as normalized distance to divide
(nDD; nDD=0 at outlet and nDD=1 at the divide; Fig. 4). Lines of isovalue of nDD considering an
increment of 0.1 are also shown in the space-time diagrams (Fig. 8). To a first order, the trajectories of
each knickpoint are very comparable within an experiment regardless the stage of retreat of the water
divide and the size of the catchment. Visually for example, in the space-time diagrams there is no
systematic variation in the general slope of the successive knickpoint trajectories over time, as the rivers
expand, that would indicate a change in mean knickpoint velocity in relation to the change in the river
length and catchment size. In detail, an inflection of trajectories is visible for many knickpoints when
they are close to the divide, for nDD > ~0.8 (Figure 8), which indicates that they slow down as they
approach the divide. The opposite is observed for some knickpoints when they are close to the outlet,
for nDD < ~0.2 / 0.3, with some trajectories suggesting, on the contrary, an acceleration after their
initiation (Fig. 8; see also Fig. 7). These qualitative interpretations are supported by the detail analysis
of retreat velocity data shown in Figure 9. For each experiment, we show in Figure 9A the stack of
successive retreat velocities of each individual knickpoint according to distance nDD. These data show
that the range of knickpoint retreat rates depends on the rate of base-level fall. Moreover, the envelopes
draw a bell-shaped distribution for each experiment, which suggests that retreat velocities are maximum
when knickpoints are located at a mid-distance between the outlet and the divide, for central values of
nDD, between 0.4 and 0.6. This is supported by summary statistics of retreat velocities at 0.1 intervals
of nDD considering all knickpoints in each experiment (Fig. 9B). Both the mean and median values
show higher rates of upstream propagation when knickpoints are in the central section of rivers in the
three experiments, and conversely lower rates near the outlet (nDD < 0.2 / 0.3) where they initiate and
start to propagate and near the divide (nDD > 0.8), as suggested by trajectories shown in Figure 8. To
further characterize this trend, we determined the position of maximum knickpoint velocity on
longitudinal profiles, hereafter $nDD_{Vmax}$, from a second order polynomial fit (Fig. 9C). $nDD_{Vmax}$ values
are very similar between experiments (0.52, 0.57 and 0.54: Table 1). They separate positive to negative
trends of knickpoint velocities versus normalized distance as also illustrated in Figure S4 (see
Supplemental Material). Data from the three experiments indicate that after their initiation near the
outlet, knickpoints first speed up with a maximum in the central part of the catchments before
decelerating near the divide. It is worth noting that this specific trend of knickpoint retreat rates is
observed regardless of the experiment stages and thus whatever the position of the divide in the box.
This applies both to rivers in the early stages of experiments evolution, i.e. when they are small as well
as for very large rivers at the end of experiments.




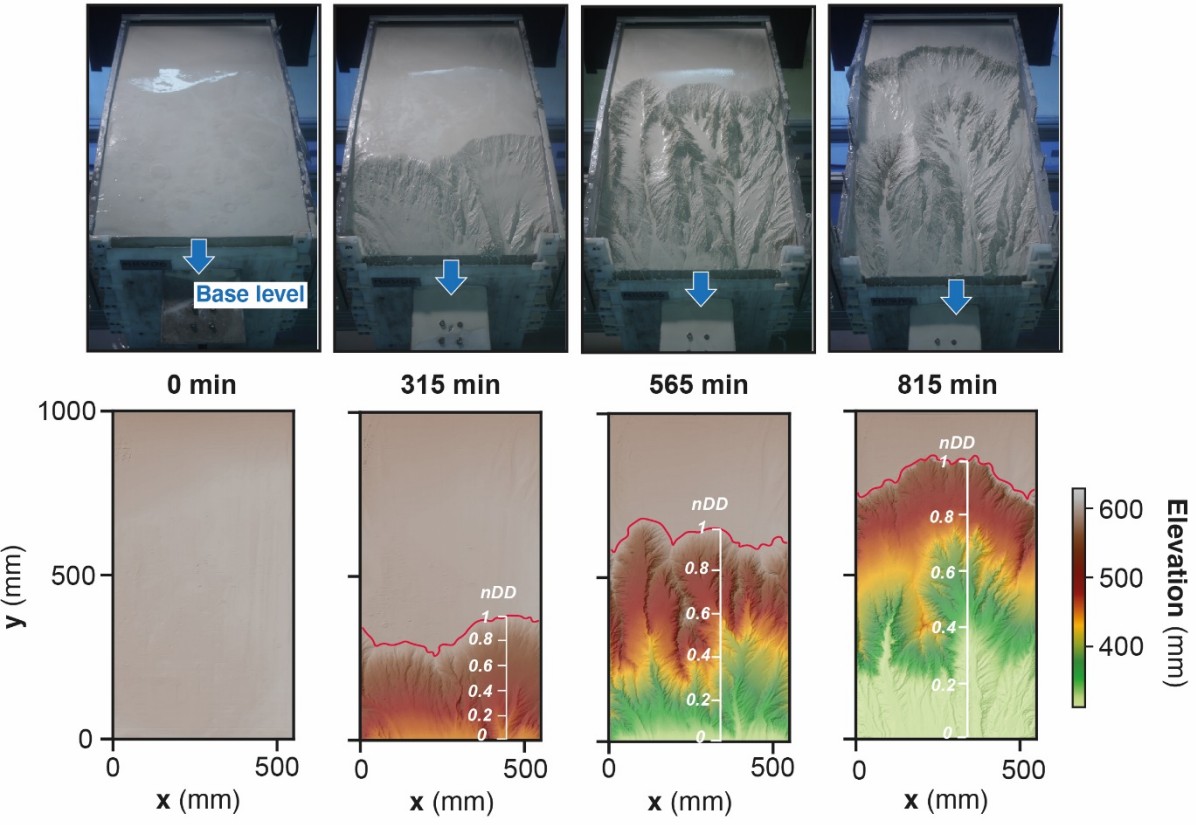

**211**

***Figure 4***. *Photos (top row) and corresponding DEMs (bottom row) of experiment BL15 at four runtimes.*

*Note the propagation of the divide (red line) through the erosion box and the drop of the sliding gate*

*used for falling base-level (blue arrows). The normalized distance to divide (nDD, see text) used to*

*follow the position of knickpoints during runs is shown superimposed to DEMs.*

**216**

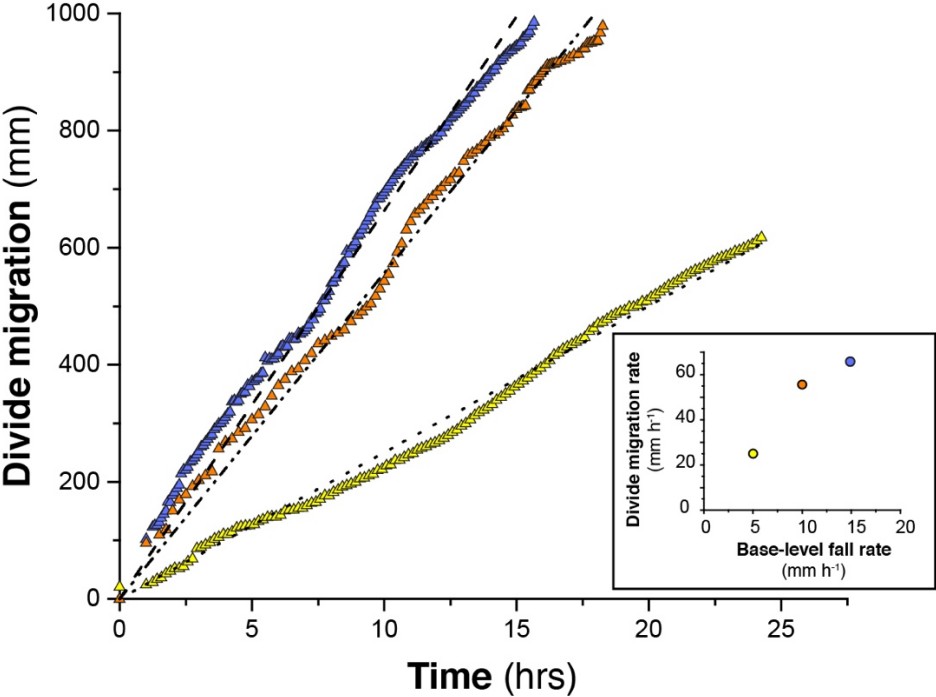

217

*Figure 5*. *Evolution of the water divide position within the erosion box for the three experiments. The inset figure (Bottom right) shows the relation between the divide migration rate in the three experiments and their related base-level fall rate.*

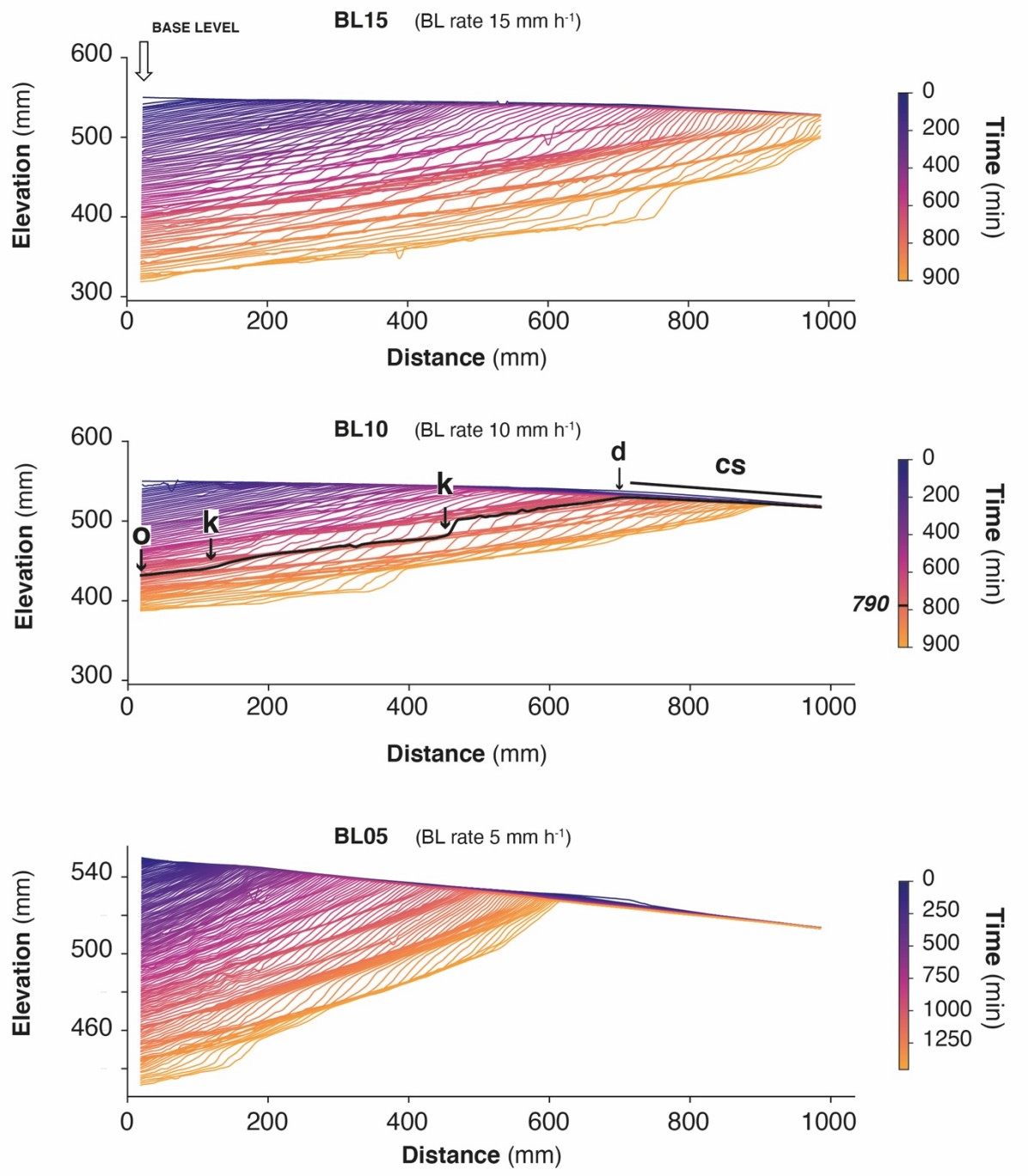

221

*Figure 6.* *Successive river longitudinal profiles of experiments, shown here every 10 min. Each longitudinal profile is colored according to experimental runtime. The sliding gate used to drop the base level is to the left. Note the initial counterslope (cs). Black thick line on BL10 is the longitudinal profile at t=790 min, illustrating the outlet (o), knickpoints (k), and water divide (d). Note the change of scale for experiment BL05.*

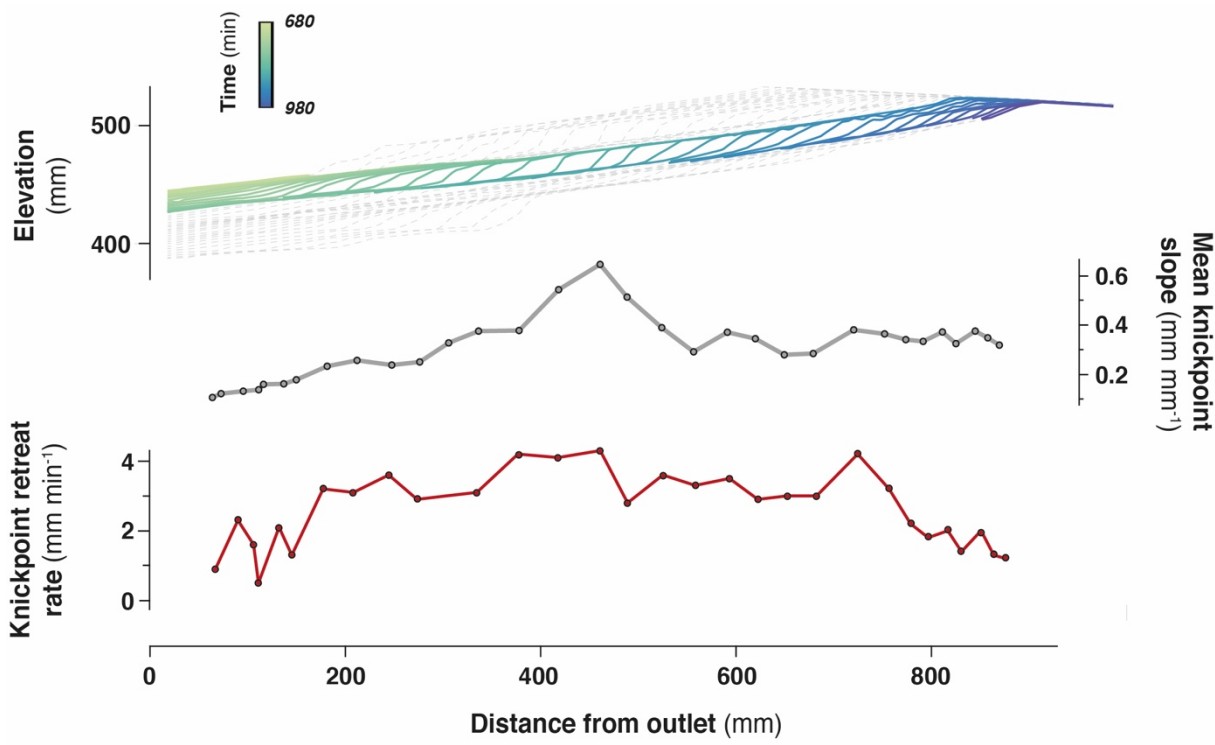

227

**Figure 7.** *Retreat of an individual knickpoint from experiment BL10 (see also Fig. 6) showing its initiation with a gentle slope which subsequently steepen as it migrates upstream (see also Fig. S3 in the Supplemental Material). Its maximum slope is reached at mid-distance between the outlet and the divide. Its lowest retreat rates are observed downstream near the outlet and upstream near the divide.*

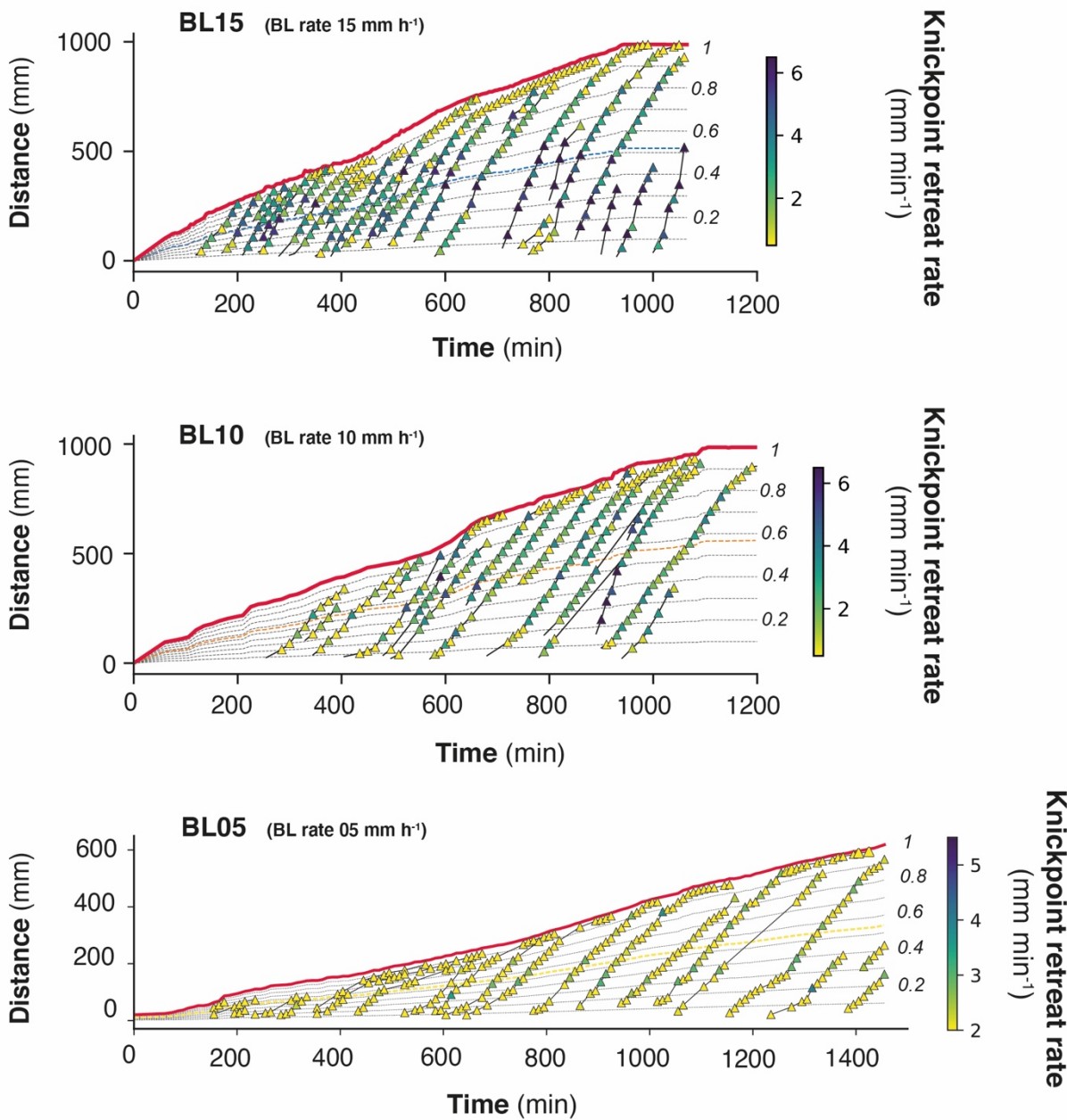

233

**Figure 8.** *Space-time diagrams showing the propagation of the water divide (red line) and successive trajectories of knickpoints (triangles). Symbols color shows instant (10 min) knickpoints retreat rate. Thin black dashed lines show the normalized distances to divide (nDD). Thin colored dashed lines show nDD$_{Vmax}$, the normalized distance where the highest rate of retreat velocity is deduced from the analysis (see text and Figure 9C). Note the change of scale and color bar for experiment BL10.*

239

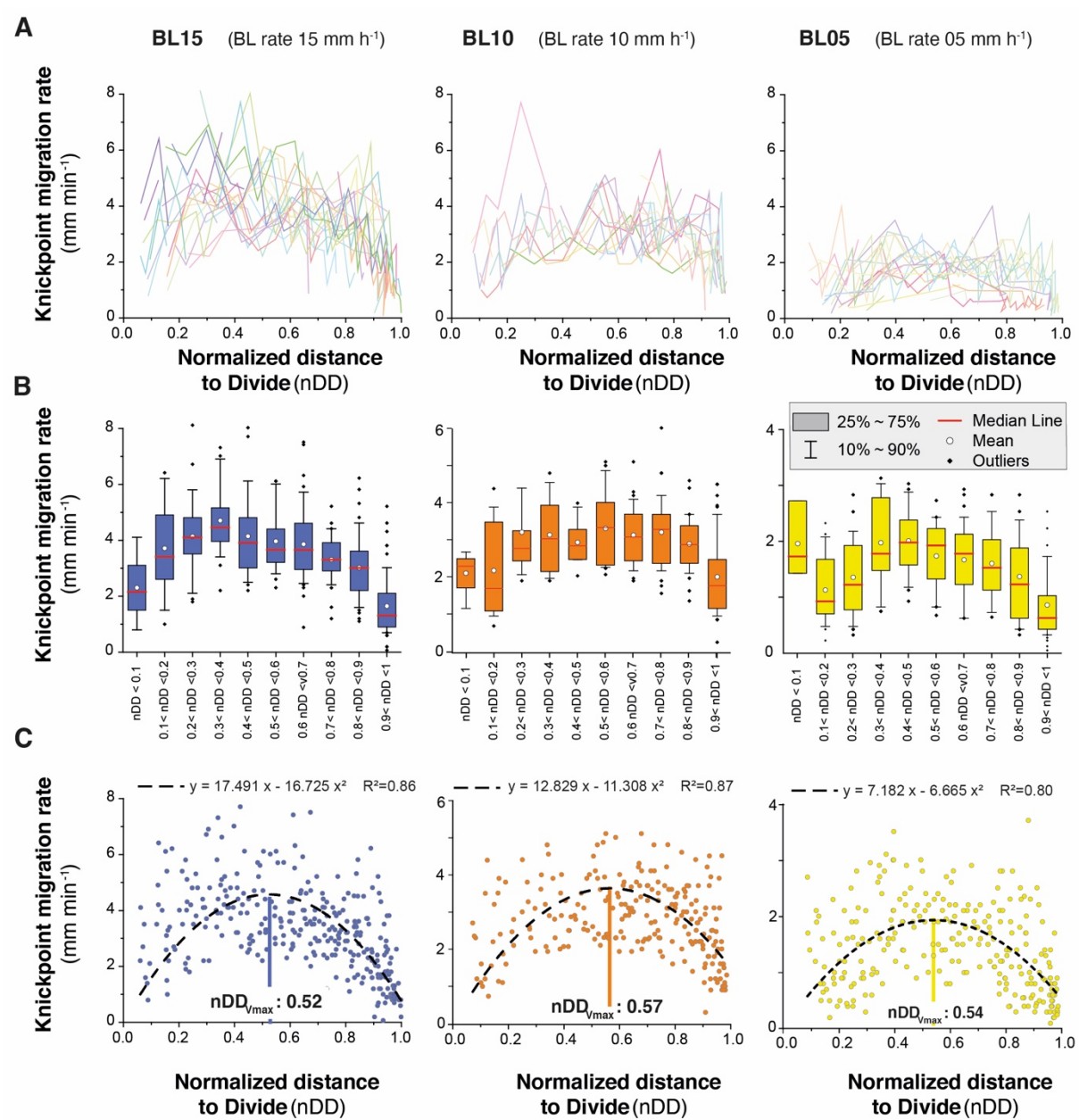

240

**Figure 9.** *(A) Knickpoint retreat rates according to the normalized distances to divide (nDD) for each knickpoint of experiments. Each color line corresponds to an individual knickpoint of the space-time diagram in Fig. 8. Note that the scale on the y-axis is the same for all graphs. (B) Summary statistics of retreat rates for nDD intervals of 0.1. Note the change in scale on the y-axis between the graphs (C) Plot of all knickpoints retreat rates for each experiment. Note the change in scale on the y-axis between the graphs. Black dashed line shows the second order polynomial fit to the data used to define the normalized longitudinal distance of maximum velocity of knickpoints (nDD$_{Vmax}$; see also Fig. S4 in the Supplemental Material).*

**3.2 Knickpoints initiation**

To illustrate how knickpoints initiated near the outlet, we consider here a 120 minute-long sequence of channel evolution in experiment BL15 during which two knickpoints (K1 and K2) successively initiate and propagate upward (Fig. 10). In addition, we analyzed the history of channel width (Fig. 11A) and unit water discharge (Fig. 11B) at a cross-section located at 8 cm from the outlet (see location on Fig. 10B). We also present a summary of the statistics of normalized elevation changes (Fig. 11C) and shear stress (Fig. 11D) for all pixels across the section. The sequence starts with a "standard" profile (i.e., a typical river profile without any perturbation) at runtimes 880 and 890 min once a previous knickpoint had already propagated through the section, still visible upstream in Figure 10A. The channel is 23 to 25 mm wide (Fig. 10B and 11A) and the unit discharge is about $1.5.10^6$ mm$^2$ h$^{-1}$. Erosion in the channel is on average lower than the base level fall as normalized erosion (erosion rate / base level fall rate) is <1 for most pixels along the section (Fig. 11C). Then, the knickpoint K1 initiates at runtime 895 min and starts to propagate upstream. At the surveyed section, the channel first narrows, up to ~15 mm wide at 905 min (~60 % decrease), and then widens (~25 mm) once the knickpoint has moved upstream of the section, at 910 min (Fig. 10B). The narrowing phase is naturally associated with an increase of the unit discharge (Fig. 11B) and with enhanced erosion greater than the base level fall rate, up to 4 times the base level fall rate in average at 900 min (Fig. 11 C), with extremes as high as 8 times the base level fall rate. Once knickpoint K1 has retreated, unit discharge decreases as the channel subsequently widens, to reach a width of 25 cm to 28 cm between 925 and 930 min (Fig. 11A) while a new regular profile, i.e. without any slope break, established at 930 min (Fig. 10A). The normalized erosion across the section decreases below the base level value (Fig. 11C), with mean erosion rate values of 0.53, 0.36 and 0.76 times below the base level rates between 915 to 925 min. Longitudinally, the profiles stack together downstream of the knickpoint following its retreat from 895 to 925 min (Fig. 10A), which also indicates minor vertical erosion here once the knickpoint has retreated despite the ongoing base level falling. The second knickpoint (K2) then initiates at 935 min, propagates upstream in a similar way, leading to the setting up of a new regular profile at 980 min downstream its position at that time (Fig. 10A). Channel narrowing is also observed on the cross-section at the passage of this second knickpoint with a width

that decreases to ~15 mm wide (Fig. 10B and 11A), associated with an increase of the unit discharge
and the erosion rate (Fig. 11C). It is followed again by a phase of widening to reach a width to around
30 / 35 mm once the knickpoint has propagated upstream and by a decreasing erosion below the base
level fall rate (Fig. 11C). Again, the longitudinal profiles stack together downstream of the knickpoint
(Fig. 10A). Note that at 975 min, most of the surveyed section is undergoing sedimentation (mean
normalized erosion rate is 0.1 and median is -0.25: Figures 10B and 11C).  The distribution of river bed
shear stress along the section is given in the Figure 11D. Despite a large variability along the section,
one can observe a significant increase of the median and maximum values at the time of the knickpoint
passage, both for K1 and K2. Once knickpoints passed, the shear stresses decrease as the river widens.
This sequence illustrates that the rivers are never in equilibrium at the 5 min time-scale, but continuously
oscillate over time between disequilibrium states with periods when channel are too wide to keep pace
with the base level, and periods of knickpoint propagation when the erosion is enhanced to catch up the
base level. The river width is the regulation parameter which allows the river erosion to adapt
by increasing or decreasing the unit discharge. These knickpoints then propagate upward up to the divide
as discussed previously (Fig. 6). The average erosion rate is similar to the base level fall rate (mean
normalized erosion rate of the sequence is 0.99) but it does not correspond to any stable configuration
of the river since the erosion rate fluctuates between smaller and larger values. Knickpoints are by-
products of this unsteady dynamics, which are generated during the phases when the river catches up
with its erosion deficit with respect to the base level.

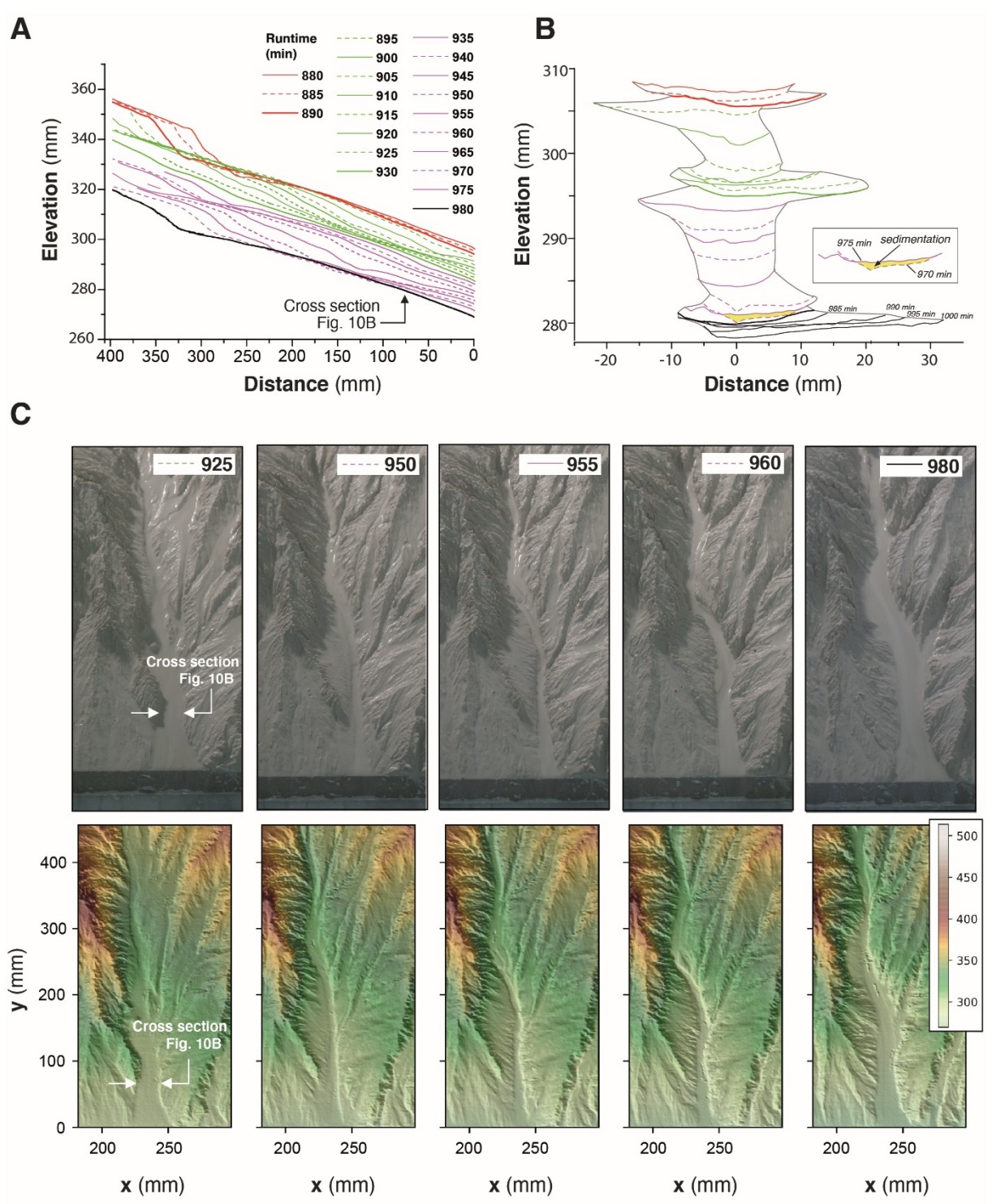

*Figure 10. Downstream knickpoints initiation and propagation in a 120 minute-long sequence of experiment BL15 from experimental runtime 880 to 1000 minutes. (A) Sequence of downstream longitudinal profiles (5 min time-interval) of the investigated river, corresponding to the sequence hydro-geomorphic parameters shown in Figures 11 and 12. Propagation of the first (K1; initiated at 895') and second (K2; initiated at 935') knickpoints is shown in green and purple colors respectively*

*(see text). (B) Time evolution of successive cross-sections of the channel at 80 mm from the outlet. Colors*
*are the same as in Fig. 10A. (C) Photos (top row) and perspective views of DEM -bottom row) at five*
*time-steps. Color bar is elevation in mm.*


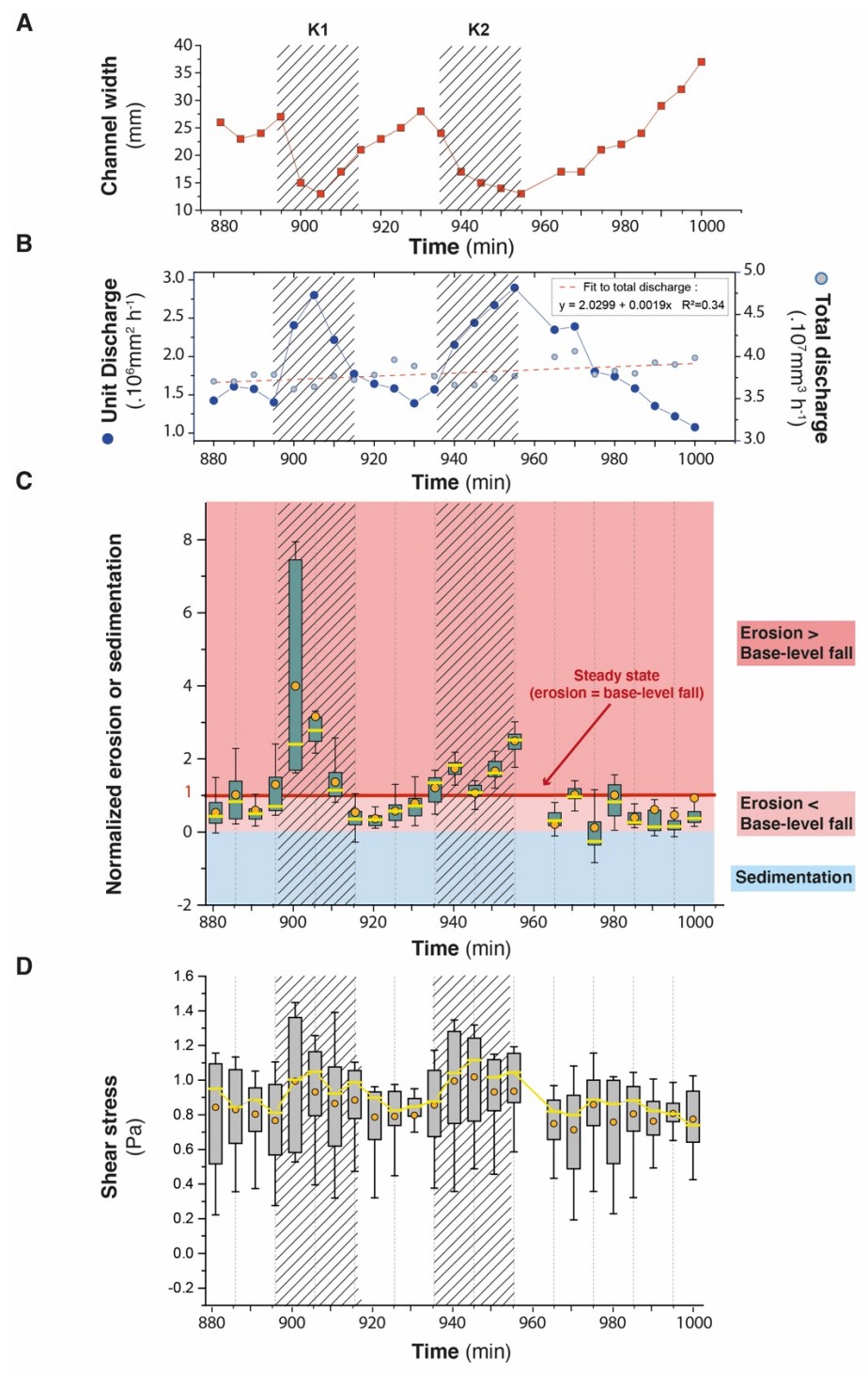


***Figure 11.*** *Time-series (5 min time interval) of river width (A) and unit and total discharge (B) for the*
*channel in experiment BL15 shown in Figure 10B (see also location of Fig. 10C). Time-series of box-*
*and-whisker plots of normalized erosion or sedimentation (C) and shear stress (D) for all pixels across*
*the channel cross-section. Orange solid circles and yellow lines show the mean and median values,*
*respectively. Edges of the boxes indicate the 25th and $75^{th}$ percentiles. Note that in C, normalized values*
*of 1 indicate erosion at the same rate as base-level fall (steady-state conditions). Values > 1 or <1*
*indicate respectively higher and lower erosion rate than BL fall rate. Negative values indicate*
*sedimentation. On all graphs, crosshatched areas indicate the passage of knickpoints K1 and K2.*
To complement cross-section data, we also illustrate (Fig. 12) how parameters vary longitudinally by
considering four stages, two before (925 min) and after (975 min) the passage of the knickpoint K2 and
two during its retreat (945 and 950 min). Note that at 925 min, the previous knickpoint (K1) has just
passed upstream the investigated profile and is responsible for the enhanced normalized erosion and
increased shear stress upstream between distance 200 to 350 mm. Similarly, at 975 min the second
knickpoint (K2) is still in the upstream part of the profile between distance 300 to 350 mm. We also
reported the longitudinal variations in river width, shear stress and normalized erosion along the profiles
(Fig. 12). At runtimes 925 and 975 min, before and after the passage of knickpoint K2, erosion is below
the base level rate along all the profiles down the knickpoints, with even localized sedimentation at 975
min between 50 and ~150 mm. These sections are characterized by low shear stress values, being
between 0.5 and 1 and by rivers that widen downward (around 0.7 mm/cm). On the opposite, during the
passage of knickpoint K2, at runtimes 945 and 950 min, mean shear stress increases locally at the
knickpoint location, being > 1 and the normalized erosion overpasses the base level rate there. These
knickpoint segments are characterized by a narrowing of the rivers as already shown previously. The
data illustrate that erosion mainly occurs during periods of knickpoint retreat though a combination of
local steepening of the profile and narrowing of the river, resulting in an increased shear stress. On the
opposite, once a knickpoint has propagated and between the passage of two successive knickpoints,
erosion decreases significantly and does not longer compensate the base level fall. These periods of
defeated erosion are characterized by low bed shear stress values in wide rivers, that widen downward.

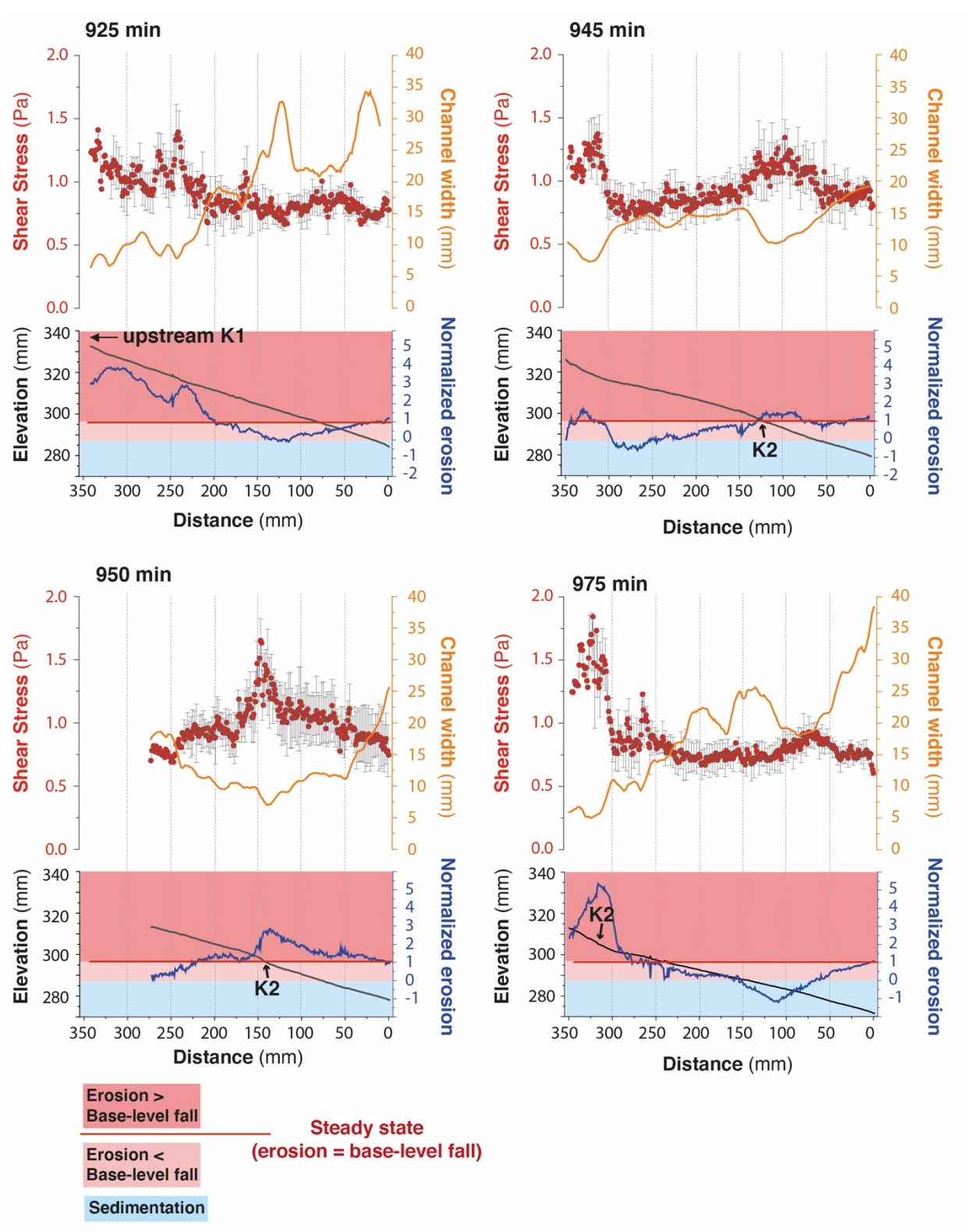

**Figure 12.** *Longitudinal trends of hydro-geomorphic parameters in experiment BL15 at runtimes 925,*

*945, 950 and 975 min (see text for comments). K1 and K2: first and second knickpoints discussed in the*

*text (see also Fig. 10A).*

**4 Discussion**

**4.1 Autogenic knickpoints**

Our experiments illustrate the generation and retreat of successive knickpoint waves that traveled across the landscape during the growth of drainage networks. They formed throughout the duration of experiments independent of the steady precipitation and base level fall rates and of the homogeneity of the eroded material. These knickpoints were autogenically generated (Hasbargen and Paola, 2000), arising only from internal geomorphic adjustments within the catchments rather than from variation in external forcing. Our observations appear very similar to those of Hasbargen and Paola (2000, 2003) and Bigi et al. (2006) who also reported the generation of successive autogenic knickpoints in landscape experiments evolving under steady forcing (rainfall and base level fall rate) throughout the duration of the runs. Unlike our experiments, which mainly consider the growth phase of drainage networks, experiments reported in Hasbargen and Paola (2000, 2003) and Bigi et al. (2006) considered the propagation of knickpoints after the phase of network growth, while their system was at steady-state on average (mean catchment erosion rate equal to base level rate). Then, given that the size of their experimental catchment was steady over time and given the steady rainfall rate, they were able to rule out variations of water discharge over time as a main driver for the generation of their knickpoints. On the opposite, in our experiments the size of catchments continuously increased over time, and thus the water discharge. However, this does not appear as a key factor controlling knickpoints initiation for several reasons. First, as we already mentioned, knickpoints arose at all stages of network growth and divide retreat, for both small and large rivers (Fig. 8), and thus whatever the range of water discharge at outlet. Second, the migration of the water divide related to drainage network growth occurred steadily and roughly at a constant rate during the experiments (see Figures 5 and 8), as well as the size of the catchments and the related increase in water discharge. Thus, we can rule out abrupt variations in discharge as the driving mechanism for knickpoint initiation. Last, knickpoint initiations occurred at a higher frequency than the increase in water discharge that resulted from catchment expansion and divide migration. For example, in addition to unit discharge, we also reported on Figure 11B the variation in total discharge during the 120 min-long sequence of knickpoint initiation discussed previously. The total

discharge rose from 3.7 $10^7$ to 4.0 $10^7$ mm$^3$ h$^{-1}$ in 120 minutes representing a ~ 8% increase, which is
relatively low compared to the ~100 % increase of unit discharge during the passage of a knickpoint.
For all these reasons we conclude that the change in catchment size was not the main driver of successive
knickpoints initiation in our experiments, which occurred at a higher frequency.
**4.2 Processes controlling knickpoints initiation and propagation**
Given that the initiation of successive knickpoints was not related to changes in external factors and
catchment size over time, we consider internal geomorphic processes as driving mechanisms. The
detailed sequence of knickpoint initiation and propagation discussed above shows enhanced incision
above the rate of base level fall during the periods of knickpoints propagation. This occurred through
local steepening of the longitudinal profile and narrowing of the river, these two factors lead to an
increase in unit discharge and bed shear stress along the knickpoints. Several studies already
documented how steepening and narrowing act together for increasing river incision rate (e.g. Lavé and
Avouac, 2001; Duvall et al., 2004; Whittaker et al., 2007; Cook et al., 2013), which is what we also
document here. The novelty in our finding here, however, lies in the evolution after knickpoint retreat.
Immediately after the retreat of a knickpoint, we show that erosion in the section of the channel where
the knickpoint just passed is inhibited despite the ongoing base level fall: river incision is lower than the
rate of base level fall, until the passage of a new knickpoint. Although only illustrated in the sequence
detailed previously (Figs. 10 to 12), this was a general behavior that occurred in all three experiments
along their whole longitudinal profile, not only their downstream part as in this sequence. This
systematic decrease in erosion downstream of the knickpoints is inherent to the geometry of the stacks
of all successive longitudinal profiles of each experiment (Fig. 6). In most cases, profiles downstream
of retreating knickpoints stack on top of each other, as illustrated schematically on Figure 13A, which
indicates minor or no erosion downstream of the knickpoints until the passage of a new one. In the case
of continuous steady adjustment of rivers to base level fall downstream of the knickpoints, the geometry
of profiles should instead show a pattern as illustrated in Figure 13B. The pattern of profiles evolution
over time documented here is usually observed following incremental drops in base level (Finnegan,
2013; Grimaud et al., 2016) and to our best knowledge this is the first time here that such geometry is
documented in the case of a continuous base level fall. This particular pattern is explained by the
decrease in erosion rate downstream of the retreating knickpoints which acts as if the base level was not
falling continuously at a constant rate but instead dropped regularly step-by-step. Therefore,
understanding the systematic occurrence of successive knickpoints in our experiments requires
understanding why erosion rate dropped downstream of knickpoints, following their retreat. After the
passage of knickpoints, we systematically observe a widening of the rivers, as also documented in
natural systems (e.g. Cook et al., 2014; Zavala-Ortiz et al., 2021) and a decrease in the bed shear stress.
Because an increase in channel width over time inevitably reduces the bed shear stress if discharge and
river gradient remain constant (Fuller et al., 2016), we propose that widening was the main factor
responsible for the decrease in shear stress and erosion rate after the passage of a knickpoint, and thus
for the occurrence of the successive autogenic knickpoints. Demonstrating the sole effect of river width
on bed shear stress and erosion rate is complicated by covariations of these factors with river slope and
variations of discharge related to connection of tributaries. This can be illustrated however on the basis
of the sequence considered previously, particularly at runtime 925 min between the passage of the two
successive knickpoints K1 and K2 (Figs. 10 and 12). At that time, the profile of the river here had a
roughly constant slope (Fig. 14), without any slope break and no major tributary connected (Fig. 10)
that could have significantly changed the water discharge. As illustrated in Figure 12, this river segment
was characterized by widening and decreasing shear stress downward despite constant slope and total
discharge. Thus, this example illustrates a decrease in shear stress that was only the result of the
widening of the river downward (Fig. 14), which supports the hypothesis that decreased erosion
downstream of the propagating knickpoints was mainly due to the widening dynamics of the
experimental rivers.

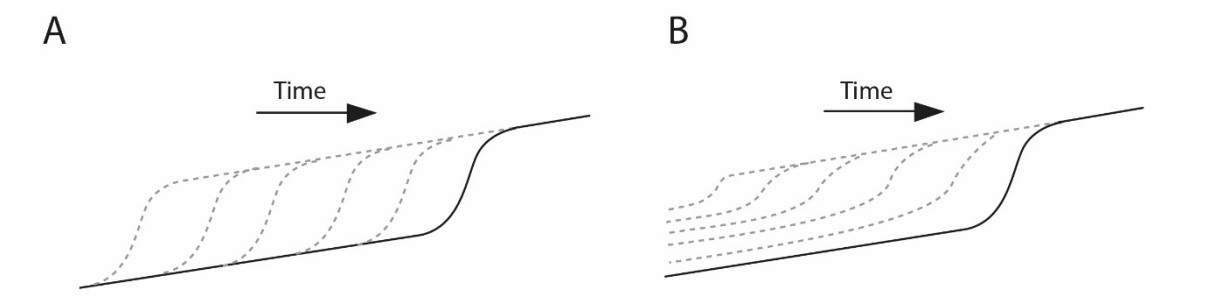


**Figure 13.** *Sketches illustrating the difference in the geometry of successive longitudinal profiles following the retreat of a knickpoint depending on whether fluvial incision is inhibited (A) or not (B) downstream of the retreating knickpoint with respect to the continuously falling base level.*

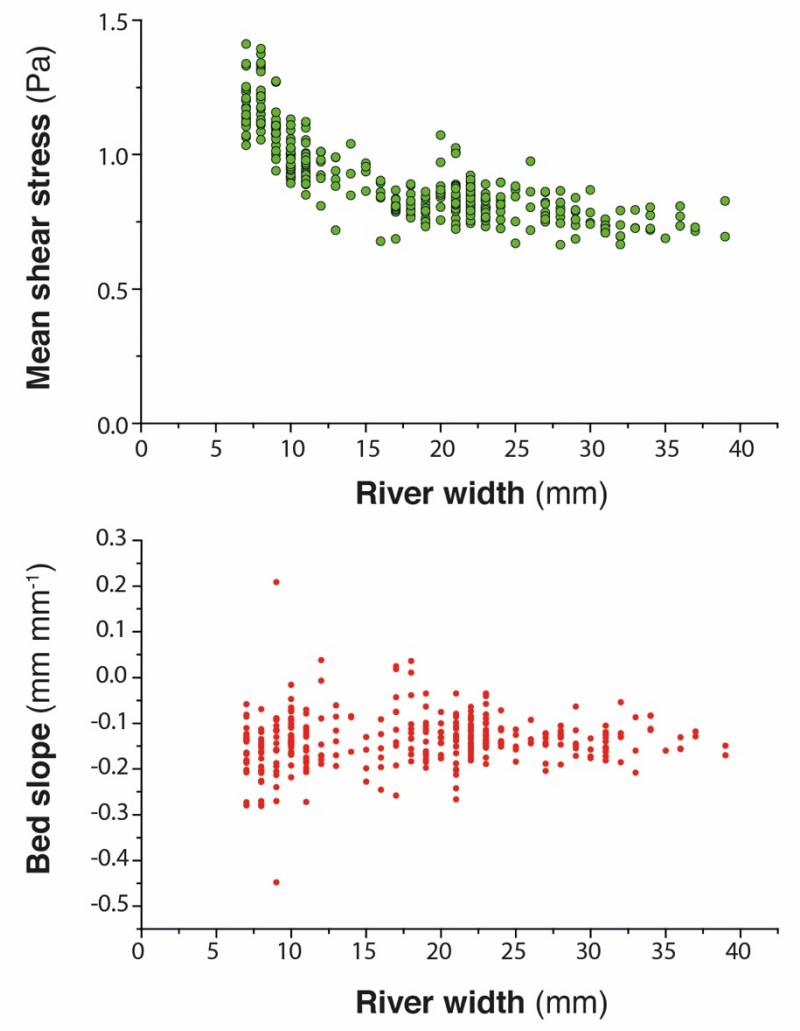

**Figure 14.** *Top: river bed shear stress versus river width in the downstream section, 40 cm-long, of experiment BL15 at runtime 925 (see also Fig. 12). Bottom: corresponding slope of the river bed.*

Incision of rivers in our experiments is fundamentally discontinuous despite continuous forcing and we highlight downstream river width dynamics, in particular river widening, as a main cause of instability. We show that once knickpoints have retreated, unit discharge, shear stress and incision rate all decrease downstream while the rivers widen, resulting in a state where incision no longer counterbalances the base-level fall. This results in an unstable situation that ends with the initiation and propagation of a new

knickpoint and a new sequence of width narrowing, increasing shear stress and incision rate, allowing
the river to recover from the incision delay accumulated during the previous widening period. Further
work is required to understand the mechanisms responsible for lateral channel erosion in our
experiments, which is a key ingredient for understanding river mobility and widening. Several field (e.g.
Hartshorn et al., 2002; Turowski et al., 2008; Fuller et al., 2009), experimental (e.g. Wickert et al., 2013;
Bufe et al., 2016; Fuller et al., 2016; Baynes et al., 2020) and numerical (e.g. Turowski et al., 2007;
Lague, 2010; Langston and Tucker, 2018; Li et al., 2021) studies have demonstrated that high sediment
flux relative to transport capacity promotes increased lateral channel erosion. Most of these studies
highlight the role of cover effect, the protection of the river bed by transient deposition of sediments on
the river bed (Sklar and Dietrich, 2001; Turowski et al., 2007, 2008; Lague, 2010; Baynes et al., 2020;
Li et al., 2021), as a main factor promoting lateral erosion in high sediment flux settings. Other studies
show that by modifying the bed roughness, sediment deposition may deflect the flow, which also
promotes lateral erosion and widening (Finnegan et al., 2007; Fuller et al., 2016). Contrary to
experimental devices specifically designed to address these issues (e.g. Finnegan et al., 2007; Fuller et
al., 2016), direct observation on actual processes that drive lateral erosion in our experiments is made
difficult by the small size of the topographic features, the depth of rivers being of millimeter scale, and
by the low grain size of the material used. Opacity due to the generation of the artificial rainfall also
considerably limits direct observation during the runs. Despite these limitations, data suggest that lateral
erosion and river widening in our experiments is also related to an increase in sediment flux. We show
that knickpoints are locations of enhanced erosion well above the rate of base level fall. We document,
for example, mean erosion rates greater than 5 times the base level fall rate, with extreme values up to
a factor of 8 locally (Fig. 11 and 12). Downstream, where rivers widen, we observe that the general
decrease in erosion rate is also associated with local deposition in some parts of the channels (for
example at runtime 915 min in Figure 11 or 975 min in Figures 10 to 12). We thus hypothesize that
lateral erosion and widening are due in part to the increase sediment flux related to enhanced erosion on
knickpoints. Further work is needed to test this hypothesis, for example by investigating in detail spatio-
temporal variations in erosion and sedimentation during width widening.
Further work is also needed to better understand how knickpoints initiate after the phases of widening,
in particular for determining whether river narrowing drives the formation of the knickpoints (e.g. Amos
and Burbank, 2007) or whether narrowing is a consequence of steepening (e.g. Finnegan et al., 2005).
Some studies that investigated river response to increased uplift rate show that narrowing alone, at
constant river gradient, can allow rivers to increase their incision rate (Lavé and Avouac, 2001; Duvall
et al., 2004; Amos et al., 2007). In this context, Amos et al. (2007) propose a model in which the river
response to an increase in uplift rate first involves width narrowing, with the increase in slope and
formation of a knickpoint occurring only in a second stage, if the increase in incision induced by
narrowing is not sufficient to counteract the uplift rate. In our experiments here, we suggest that channel
narrowing predates, and in fact enables, the steepening of the profile in the initial stages of knickpoints
formation. Indeed, we observe that the transition from a wide to a narrow channel occurs very quickly,
at a smaller time scale than the time interval between two successive digitization of the experiments (5
min), and the knickpoints that form then have a very gentle slope, which then amplifies as they migrate
upstream (Fig. 7). This suggests that it is not the steepening that drives river narrowing but on the
contrary that narrowing is essential for knickpoints to initiate. Further work would also be needed to
verify this hypothesis, in particular with additional experiments with much higher frequency of data
acquisition to capture these changes in much more detail.
**4.3 Implications**
Knickpoints in river longitudinal profiles are commonly related to variations in tectonics or climate
through their influence on base level and/or sediment supply (e.g. Whipple and Tucker, 1999; Crosby
and Whipple, 2006; Kirby and Whipple, 2012; Whittaker and Boulton, 2012) and are then used to
highlight such changes when interpreting their occurrence in natural systems. The recognition here that
knickpoints may be generated autogenically due to cycles of river widening and narrowing is then of
first importance for retrieving information on tectonics and climate from their record in landscapes in
the form of knickpoints. Finding criteria that could be used in the analysis of natural systems to
differentiate these autocyclic knickpoints from those formed in response to tectonics or climate would
be an important step in the continuation of this work. A specificity of knickpoints in our experiments is

to initiate downstream with a gentle slope, which subsequently steepen in the early stages of migration, and as a hypothesis we suggest that this may be characteristic of their autogenic formation following the mechanism described here. Being able to recognize these autogenic knickpoints would also be important for studies that investigate knickpoints propagation (e.g. Crosby and Whipple 2006; Berlin and Anderson, 2007; Schwanghart and Scherler, 2020) because knickpoints in our experiments are characterized by an upward dynamic of retreat that is not conventional. According to stream-power based celerity models, these studies consider that the upstream propagation rate of knickpoints depends inversely on drainage area (a proxy for discharge; Crosby and Whipple 2006; Berlin and Anderson, 2007), implying a monotonous decrease of their retreat rate as they propagate upstream due to the progressive reduction of drainage area and water discharge. This property is used for example to invert their present location for dating the external perturbation responsible for their formation (Crosby and Whipple 2006; Berlin and Anderson, 2007). Here, knickpoints in our experiments first accelerate during their initial stages of propagation before decelerating in a second time as they approach the divide (Fig.9). Only this later phase of decreasing knickpoint velocity in the upstream part of rivers (for normalized distance NDD > $nDD_{Vmax}$: Fig. 9) is consistent with predictions from stream-power based celerity models (see Fig. S5 in the Supplemental Material). On the opposite, a sole control by drainage area and discharge cannot explain the increase in velocity observed in the downstream sections (for NDD < $nDD_{Vmax}$: Fig. 9), which implies an additional controlling factor. We suggest that this specific mode of retreat downstream is related to the progressive steepening of the knickpoints rather than to a purely hydrologic control. Deciphering the respective roles of slope and discharge in the retreat dynamics documented would require further in-depth analysis, particularly during the early stages of initiation and propagation which appear to be specific to the autogenic mechanism defined here.

We show that the formation of knickpoints in our experiments is closely related to periods of decreasing erosion rate as the rivers widen, counterbalanced by increasing rate greater than the rate of base level fall as the rivers narrow and knickpoints form. Thus, the sequential evolution of longitudinal profiles is very similar to the geometry that would be observed if the system was forced by discrete drops of the base level, rather than by a continuous drop as it is actually the case. We did not measure the sediment

flux at the output of our models, but we can assume that it would be characterized by fluctuations
controlled by the frequency of knickpoint initiation, superimposed on a longer-term increasing trend
related to the growth of drainage networks. Some sediment outflux fluctuations were actually measured
by Hasbargen and Paola (2000) in their experiments and interpreted as the consequence of knickpoint
propagation. This study and our work illustrate that fluctuations in sediment flux can be observed at
catchments outlet despite constant forcing parameters, when autocyclic knickpoints are generated in
river systems.
By performing such exploratory experiments, we do not pretend to reproduce natural landscapes in the
laboratory because of important scaling issues (see Paola et al., 2009 for an extensive reflection on this
matter) but rather to highlight and document complex system behaviors under controlled conditions that
could provoke further investigations. Our findings support ongoing investigations that aim in better
understanding the links between lateral erosion, channel geometry and valley width which is an issue
that is emerging in the last years (e.g. Turowski, 2018; Croissant et al., 2019; Langston and Tucker,
2019; Baynes et al., 2020; Zavala-Ortiz et al., 2021). A perspective to our work would be to investigate
the mechanism of knickpoints generation driven by river width variations and the conditions that lead
to their formation using landscape evolution models that incorporate lateral erosion and a dynamic river
width (e.g. Davy et al., 2017; Carretier et al., 2018; Langston and Tucker, 2019). Simulations of
Langston and Tucker (2019) highlight the role of bedrock erodibility as an important factor controlling
lateral migration of rivers and the width of valleys, an issue that has not been investigated here given
the similarity of the eroded materials in our experiments here. This study also confirms the assumption
of Hancock and Anderson (2002) that lateral erosion and widening occurs preferentially in contexts of
low incision rate, *i.e.* in domains with low uplift rate. This is likely in such contexts that the new mode
of autogenic knickpoints formation driven by river width dynamics that we define in this study should
apply.
**5 Conclusion**
Knickpoints in the longitudinal profile of rivers are commonly assumed to be incisional waves that
propagate upstream through landscapes in response to changes in tectonics, climate or base-level. Based

on results from a set of laboratory experiments at the drainage basin scale that simulate the growth of drainage networks in response to constant base level fall and rainfall, we show that knickpoints also form autogenically, independent of any variations in these external forcing factors. In all experiments, successive knickpoints initiate and propagate upward throughout the duration of the experimental runs, independent of the rate of base level fall applied and of the size of the rivers as the catchments expand. Thanks to the computation of hydraulic information (water depth, river width, discharge and shear stress) using a hydrodynamic model, we show that the formation of knickpoints is driven by variations in river width at the outlet of catchments and we highlight width widening as a main cause of instability leading to knickpoint formation. Widening entails a decrease in shear stress and an incision rate lower than the rate of base level fall, resulting in an unstable situation that ends up with a sequence of width narrowing, increasing shear stress and incision rate as a knickpoint initiates. Rivers in our experiments thus evolve following sequences of width widening and narrowing that drive the initiation and propagation of successive knickpoints. As a result, incision is fundamentally discontinuous over time despite continuous forcing. It occurs during discrete events of knickpoint propagation that allow the rivers to recover from the incision delay accumulated during widening periods.

**Author contributions.** SB designed the experimental device. LdL, SB and AG built the experimental setup and carried out the experiments. LdL analyzed the data with the help of SB and PhD. All authors discussed the data. LdL and SB wrote the manuscript with input from AG and PhD.

**Acknowledgements.** This work was supported by ORANO-Malvesi and CNRS-INSU Tellus-Syster programme. We thank Sebastien Carretier and Odin Marc for fruitful discussions and Jens Turowski for his comments on a preliminary version of this manuscript. We thank Laure Guerit and an anonymous reviewer for their constructive comments which greatly improved the manuscript.

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
