# Peer review of "Léopold de Lavaissière1, Stéphane Bonnet1, Anne Guyez1, and Philippe Davy2"

_Earth Surface Dynamics, 2021_

## Author Comment (AC1)

**Reply to comments by referee #1**

**RC1**: 'Comment on esurf-2021-50', **Anonymous Referee #1,**

**In the submitted manuscript Lavaissière and colleagues present new laboratory experiments documenting the generation and retreat of autogenic knickpoints. Overall, I am supportive of this manuscript (the experiments are really cool!) and believe that it should be eventually published in ESurf. Autogenic knickpoints are seeing increasing study in our field, and experiments, like those detailed in the submitted manuscript, are important for advancing our knowledge of how these features form, what controls their retreat rate, and how we might separate autogenic dynamics from climate/tectonic forcing in natural landscapes. The submitted manuscript has potential to address all these issues. I believe that with some moderate revisions, the authors can revise the manuscript to be both more impactful in our community, and easier for readers to digest. I've outlined my comments below, and offer the comments as constructive criticism for a manuscript that I think is cool, an important contribution, and which I look forward to seeing eventually published.**

We thank the reviewer for his thorough review and are glad he recognizes the value in our physical experimental approach. We have responded to each of his detailed comments below.

**1. Mechanism of knickpoint formation and retreat**

**I found the manuscript to be very detailed on observations, which is good, as the authors did an excellent job of relating the results of their experiments. However, I believe the manuscript could be more impactful if the authors expanded their discussion of the mechanisms that control both autogenic knickpoint formation and knickpoint retreat.**

We have followed the reviewer's recommendation and significantly expanded the discussion of formation and retreat mechanisms (see below). For this we have also added a new figure (now **Figure 7**) that comes in support of the additional explanations that we have added in the manuscript.

**For knickpoint formation, the abstract stresses channel narrowing as the initiation mechanism, but it seems to me after reading the manuscript that knickpoint formation may actually cause channel narrowing (i.e., the narrowing occurs after knickpoint initiation) in the experiments. Can the authors use their experimental data to separate the role of channel narrowing and channel steepening in knickpoint creation?**

This is a tricky question that comes up several times in the following comments. We now specifically addressed this issue in the discussion (section 4.2, now lines 454 to 470):

*"Further work is also needed to better understand how knickpoints initiate after the phases of widening, in particular for determining whether river narrowing drives the formation of the knickpoints (e.g. Amos and Burbank, 2007) or whether narrowing is a consequence of steepening (e.g. Finnegan et al., 2005). Some studies that investigated the rivers response to increased uplift rate show that narrowing alone, at constant river gradient, can allow rivers to increase their incision rate (Lavé and Avouac, 2001; Duvall et al., 2004; Amos et al., 2007). In this context, Amos et al. (2007) propose a model in which the river response to an increase in uplift rate first involves width narrowing, with the increase in slope and formation of a knickpoint occurring only in a second stage, if the increase in incision induced by narrowing is not sufficient to counteract the uplift rate. In our experiments here,*

*we suggest that channel narrowing predates, and in fact enables, the steepening of the profile in the initial stages of knickpoints formation. Indeed, we observe that the transition from a wide to a narrow channel occurs very quickly, at a smaller time scale than the time interval between two successive digitization of the experiments (5 min), and the knickpoints that form then have a very gentle slope, which then amplifies as they migrate upstream (Fig. 7). This suggests that it is not the steepening that drives river narrowing but on the contrary that narrowing is essential for knickpoints to initiate. Further work would also be needed to verify this hypothesis, in particular with additional experiments with much higher frequency of data acquisition to capture these changes in much more detail."*

**Based on my reading of the manuscript and looking at the data, I think that, after the initial autogenic knickpoint has been created, spatially variable erosion caused by the initial knickpoint formation leads to channel steepening, which then allows additional knickpoint formation and channel narrowing. Is this correct? If so, it would be helpful to make this clear in the manuscript.**

Yes, that's about right. We show that after the passage of a knickpoint (which correspond to a steepening of the river), widening leads to a fall of erosion rate and a river which is in disequilibrium, what the reviewer qualifies as "spatially variable erosion". This state of disequilibrium subsequently ends up with the formation of a new knickpoint. This was already explained in the discussion section in the previous version of the manuscript (lines 422-427 of the former ms); we rephrased it as (now lines 423-427):

*"We show that once knickpoints have retreated, unit discharge, shear stress and incision rate all decrease downstream while the rivers widen, resulting in a state where incision no longer counterbalance the base-level fall. This results in an unstable situation that ends up with the initiation and propagation of a new knickpoint and a new sequence of width narrowing, increasing shear stress and incision rate, allowing the river to recover from the incision delay accumulated during the previous widening period. "*

**If not, it would be helpful to document that channel narrowing occurs before knickpoint creation if it is indeed the change in channel width that controls knickpoint formation. I recommend the authors create a schematic diagram/cartoon to include in the manuscript that shows the steps leading to knickpoint formation.**

We actually suggest that width narrowing occurs before steepening as explained in our response above and now discussed in the revised manuscript (e. g. lines 467-468: *"This suggests that it is not the steepening that drives river narrowing but on the contrary that narrowing is essential for knickpoints to initiate."*). Note that we did not succeed to create a satisfying cartoon that was clear enough to explain all the details of the mechanism of knickpoints generation. Sorry !

**I think I understand the authors' mechanism for how autogenic knickpoints continue to be created once the initial one forms (spatially-variable erosion due to changing channel width result in changes in river profile concavity). However, what creates the initial knickpoint? Can this be determined from the experimental data? A discussion of this would be helpful.**

We don't have any evidence that the very initial knickpoint forms due to a different mechanism as the successive ones. The first knickpoint does not form at the very beginning of the experiments but only after water flow starts to localize to form a first small drainage network but this is a subject we have not studied in depth.

**Sedimentation was mentioned briefly in the manuscript, and sedimentation has been shown to play an important role in knickpoint formation and retreat in previous studies (e.g., Grimaud et al 2016; Scheingross et al., 2019). It was unclear to me if**

**sedimentation and cover of the bed contributed to the formation of knickpoints in these experiments. Discussing this in more detail would be helpful.**

We actually sometime observed that sedimentation occurs downstream a retreating knickpoint where the river widens and shear stress decreases. This is illustrated in 10B, 11C and 12) and already mentioned in the text (now lines 284 and 324, formerly lines 282 and 322 in the initial ms). However, in most cases documented here widening downstream a retreating knickpoint leads to a decrease in erosion rate (below the rate of base-level all) and this is only in some extreme cases that it goes with a change from erosion to sedimentation along all the river cross-sections. Given the focus of our paper on widening and shear stress, we cannot document this process in detail as done by Grimaud et al. (2016) for example, this is out of the scope of the paper here, but we agree that this is an issue that will deserve investigation.

**For knickpoint retreat, what is the mechanism that drives this retreat? Are knickpoints discrete steps that are undercut (e.g., Baynes et al., 2018)? Or is it simply the steeper slope of the knickpoint leads to increased erosion rate relative to the surrounding?**

Some of the steepest knickpoints actually sometimes show undercutting but it is quite rare. The main mechanism that drives their retreat is indeed related to the steeper slope.

**I also had questions about the controls on knickpoint retreat rate. The authors argue that knickpoint retreat rates follow a 'bell shaped curve' where retreat rates are initially slow, rates increase up until the knickpoint has gone approximately half way to the divide, and then rates slow. I have two questions about this:**

**First, I think this pattern seems clear in experiment MBV06; however, the pattern is less clear to me in experiments MBV07 and MBV09. If the authors want to argue for this pattern in their experiments, I would encourage them to perform some statistical tests to show that the quadratic they fit to the data in Fig. 8C is indeed the most appropriate description of the data. For example, can the authors show that the observed patterns of retreat rate vs. time are statistically distinct from a constant retreat rate? My guess (from looking at Fig. 8C) is that the retreat rate vs. time data for experiment MBV09 might be equally well described by a constant retreat rate. If this is the case, this raises a new an important observation that base level drop rate can play an important role in setting how retreat rate varies with distance upstream.**

Regarding the first comment, we agree that the bell shaped curve for experiment MBV09 shown in former Figure 8C (now 9C) is not as clear as for the other two experiments. Note however that we decided to keep the same scale for y-axis in the three graphs in order to illustrate the dependency of knickpoint retreat rates to the base-level fall rate. Moreover, statistical data shown in Figure 8B (now 9B; mean and median values) also support a higher knickpoint migration rate for nDD ~0.5 / 0.6 even for experiment mBV09. We added these explanations in the revised ms (lines 197-202):

*"Note that because knickpoint retreat rates also depend on the rate of base-level fall, the range of retreat rates is smaller in experiment with the lower rate of base level fall, BL05, so that their variation with distance is not as well defined as in both other experiments. However, the mean and median values are also slightly higher for intermediate distances which suggests that the trends described for the other two experiments are also valid here."*

**Second, perhaps I missed it, but I didn't see a mechanistic explanation of why the retreat rate would follow a bell shaped curve. I would encourage the authors to provide a detailed mechanistic description of why this should occur.**

Yes, we fully agree, it was a lack in the former ms. We now specifically discussed this issue in the ms (section 4.3). Note that we added a new figure (Now Fig. 7) that comes in support to our explanation for the bell shaped curve (lines 485-502 of revised ms):

*"... knickpoints in our experiments are characterized by an upward dynamic of retreat that is not conventional. According to stream-power based celerity models, these studies consider that the upstream propagation rate of knickpoints depends inversely on drainage area (a proxy for discharge; Crosby and Whipple 2006; Berlin and Anderson, 2007), implying a monotonous decrease of their retreat rate as they propagate upstream due to the progressive reduction of drainage area and water discharge. This property is used for example to invert their present location for dating the external perturbation responsible for their formation (Crosby and Whipple 2006; Berlin and Anderson, 2007). Here, knickpoints in our experiments first accelerate during their initial stages of propagation before decelerating in a second time as they approach the divide (Fig.9). Only this later phase of decreasing knickpoint velocity in the upstream part of rivers (for normalized distance $NDD > nDD_{Vmax}$: Fig. 9) is consistent with predictions from stream-power based celerity models (see Fig. S3 in the Supplemental Material). On the opposite, a sole control by drainage area and discharge cannot explain the increase in velocity observed in the downstream sections (for $NDD < nDD_{Vmax}$: Fig. 9), which implies an additional controlling factor. We suggest that this specific mode of retreat downstream is related to the progressive steepening of the knickpoints (Fig. 7) rather than to a purely hydrologic control. Deciphering the respective roles of slope and discharge in the retreat dynamics documented would require further in-depth analysis, particularly during the early stages of initiation and propagation which appear to be specific to the autogenic mechanism defined here. "*

**Finally, for both knickpoint formation and retreat, I would encourage the authors to discuss if the material used in the experiments imparts any bias on the results. My understanding is that the silica paste used in these experiments erodes via clear water flow, unlike natural rock which (in many cases) erodes primarily via abrasion from sediment impacts. Would you expect different results in a natural landscape where erosion occurs primarily via abrasion? Some discussion of this would strengthen the manuscript.**

As we did not perform experiments with different materials, a discussion on this topic would be suggestive. The reader could refer to other publications that specifically address this issue (e.g. Bonnet and Crave, 2006, see ref in the ms; Graveleau et al., Tectonophysics, 513, 68-87, 2011; Baynes et al., 2018, see ref in the ms; Reitano et al., Earth Surf. Dynam., 8, 973-993, 2020).

We however added the following sentence in the discussion (Lines 524-527): "*Simulations of Langston and Tucker (2019) highlight the role of bedrock erodibility as an important factor controlling lateral migration of rivers and the width of valleys, an issue that has not been investigated here given the similarity of the eroded materials in our experiments here*." Note that erosion in our experiments does not occur via clear water flow but through highly concentrated flow.

**2. Expand the discussion on the implications of the work**

**The authors motivate their study with arguments that knickpoints are often used to read tectonic and climate history recorded in river profiles. While I think the authors do a nice job of showing how knickpoints can form autogenically, there's no discussion about what the implications of this are for inverting river profiles or for landscape evolution. In order to make their manuscript more impactful, I encourage the authors to expand their discussion of the implications of their work. For example, how can we use the experimental results to help separate autogenic versus tectonically (or climatically or lithologically) created knickpoints in the field? Do the autogenic knickpoints created in the experiment have any characteristic scales or morphology that one could look for in the field to identify them?**

We agree with the reviewers that many issues would deserve to be discussed and therefore investigated thanks to the new and unique device that we used here. This new device offers for the first time the possibility to investigate the geomorphic dynamics at the scale of a large drainage basin, 1 meter-long, under controlled conditions. We show here a complex dynamic at the catchment-scale that would have been difficult, if not impossible, to document using a device at the reach-scale for example. We made significant efforts to monitor and document a new mechanism of autogenic knickpoint generation and a next step would obviously be to find for criteria that could be used to help separate them from those generated in response to tectonic or climatic changes. A first step for that would be to perform new experiments with such changes in the control parameters, which would definitely provide new insights useful for comparison. As a hypothesis, we suggest that the progressive steepening of the knickpoints that we observe in the downstream part is diagnostic to their autogenic formation as described here. Actually, we can assume that a knickpoint generated in a response to faulting or increasing uplift rate would be primarily steep and would eventually propagate with a constant or declining slope. Indeed, to our knowledge the steepening observed here as never been observed before.

We added a new section in the discussion (4.3 Implications) that address specifically the implications of our finding:

Lines 475-514 : "*... The recognition here that knickpoints may be generated autogenically due to cycles of river widening and narrowing is then of first importance for retrieving information on tectonics and climate from their record in landscapes in the form of knickpoints. Finding criteria that could be used in the analysis of natural systems to differentiate these autocyclic knickpoints from those formed in response to tectonics or climate would be an important step in the continuation of this work. A specificity of knickpoints in our experiments is to initiate downstream with a gentle slope, which then amplifies in the early stages of migration, and as a hypothesis we suggest that this may be characteristic of their autogenic formation following the mechanism described here. Being able to recognize these autogenic knickpoints would also be important for studies that investigate knickpoints propagation (e.g. Crosby and Whipple 2006; Berlin and Anderson, 2007; Schwanghart and Scherler, 2020) because knickpoints in our experiments are characterized by an upward dynamic of retreat that is not conventional. According to stream-power based celerity models, these studies consider that the upstream propagation rate of knickpoints depends inversely on drainage area (a proxy for discharge; Crosby and Whipple 2006; Berlin and Anderson, 2007), implying a monotonous decrease of their retreat rate as they propagate upstream due to the progressive reduction of drainage area and water discharge. This property is used for example to invert their present location for dating the external perturbation responsible for their formation (Crosby and Whipple 2006; Berlin and Anderson, 2007). Here, knickpoints in our experiments first accelerate during their initial stages of propagation before decelerating in a second time as they approach the divide (Fig.9). Only this later phase of decreasing knickpoint velocity in the upstream part of rivers (for normalized distance $NDD > nDD_{Vmax}$: Fig. 9) is consistent with predictions from stream-power based celerity models (see Fig. S3 in the Supplemental Material). On the opposite, a sole control by drainage area and discharge cannot explain the increase in velocity observed in the downstream sections (for $NDD < nDD_{Vmax}$: Fig. 9), which implies an additional controlling factor. We suggest that this specific mode of retreat downstream is related to the progressive steepening of the knickpoints (Fig. 7) rather than to a purely hydrologic control. Deciphering the respective roles of slope and discharge in the retreat dynamics documented would require further in-depth analysis, particularly during the early stages of initiation and propagation which appear to be specific to the autogenic mechanism defined here.*

*We show that the formation of knickpoints in our experiments is closely related to periods of decreasing erosion rate as the rivers widen, counterbalanced by increasing rate greater than the rate of base level fall as the rivers narrow and knickpoints form. Thus, the sequential evolution of longitudinal profiles is more consistent with the geometry that would be observed if the system was forced by discrete drop of the base level, rather than by a continuous base level drop as it is actually the case. We did not measure the sediment flux at the output of our models, but we can assume that it would be characterized by fluctuations controlled by the frequency of knickpoint initiation,*

*superimposed on a longer-term increasing trend related to the growth of drainage networks. Some sediment outflux fluctuations were actually measured by Hasbargen and Paola (2000) in their experiments and interpreted as the consequence of knickpoint propagation. This study and our work illustrate that fluctuations in sediment flux can be observed at catchments outlet despite constant forcing parameters, when autocyclic knickpoints are generated in river systems."*

**Are there certain landscapes or lithologies that will be more prone to autogenic knickpoint formation than others?**

It also an interesting issue that we have also briefly addressed at the end of the new ms by adding the following sentences:

Lines 524-531: "*Simulations of Langston and Tucker (2019) highlight the role of bedrock erodibility as an important factor controlling lateral migration of rivers and the width of valleys, an issue that has not been investigated here given the similarity of the eroded materials in our experiments here. This study also confirms the assumption of Hancock and Anderson (2002) that lateral erosion and widening occurs preferentially in contexts of low incision rate, i.e. in domains with low uplift rate. This is likely in such contexts that the new mode of autogenic knickpoints formation driven by river width dynamics that we define in this study should apply.*"

**3. Motivation of the problem and knowledge gap in the introduction**

**The introduction did not seem to state a clear problem, question or knowledge gap which the authors were addressing in this study. Instead, this comes in the methods, when the authors describe how previous experiments used a fixed outlet while in this experiment the rivers are free to adjust their width at the outlet. This is an important point that shows how the contribution here is unique relative to previous work. I suggest the authors revise the introduction to add this information and further expand upon a statement of the overall question and knowledge gap would (in my opinion) strength the paper.**

We agree with the reviewer and moved the sentences regarding the specificity of the design that were initially in the method section to the introduction.

Lines 60-66: "*The experiments presented here have been performed using a new setup specifically designed to investigate the evolution of a large, meter-long, single drainage basin under controlled forcing condition. In previous similar catchment-scale experiments (Hasbargen and Paola, 2000, 2003; Bigi et al., 2006; Rohais et al., 2012) the outlet location was pinned to a narrow motor-controlled gate used to simulate base-level fall and which also set the river width at the outlet. A specificity of our setup here is to use a large gate instead of a narrow one, allowing experimental rivers to freely evolved downstream, with no constraints on their width.*"

**4. Writing style**

**Throughout the results and discussion session, I found the authors spent significant time describing the figures with detail in the manuscript, instead of placing this information in figure captions. Especially when reading the results, it felt to me as if the authors were describing the figures, rather than trying to describe the results of the experiments and the processes that were observed.**

**I think the length of the paper could be reduced, and the paper would be easier to read, if the authors tried to re-write these sections to focus more on the observations and results themselves, and saved the details of the figures for the figure caption. In practice, this could be accomplished by a switch in writing style. In place of text such**

as "In Fig. X, we show this and that, and these observations indicate this process" the authors could instead write "We observed this process (Fig. X)" and then include additional details of what's plotted in the figure caption. For example, L255-260 is all text that I think belongs in the figure caption and not in the main manuscript. This is just one example, but this occurs throughout the manuscript. I think that streamlining this text will free additional space for the authors to discuss some of the issues I've raised above as well as comments that may come from the other reviewers.

We agree and rephrased the ms accordingly.

Related, there were a handful of English language errors throughout the manuscript. I've pointed out a few of them below, but not all. These errors did not affect my ability to read and comprehend the manuscript, and were rather minor, but should be fixed.

**Minor and Line-by-line comments**

Somewhere in the introduction or methods, it may be useful to distinguish a knickpoint from a step in the profile. Tectonic geomorphologists are often interested in knickpoints that extend hundreds of meters to kilometers across long profiles, whereas steps can form at much smaller spatial scales in channels. Both steps and these longer length steepened channel reaches have been called knickpoints in the literature. Which features are the authors referring to in their use of the term 'knickpoint'? And are the knickpoints generated in the experiment better described as individual steps or steepened channel reaches that extend over a significant portion of the channel length? I think it's the latter, but this should be made clear for readers.

We consider here changes in river gradient that may appear as knickzones or as knickpoints and that are associated with drops in erosion rates. We now highlight that their shape may change through time by adding a new figure (now Fig. 7) and by adding the following sentences:

Lines 159-161: "*The successive longitudinal profiles of the main river investigated in each experiment (Fig. 6) illustrate the growth of rivers as they propagate within the box. These profiles show alternations of segments with low and higher slopes, the later defining knickpoints*"

Lines 164-168: "*A characteristic of these knickpoints highlighted in Figure 7 (see also Fig. 6) is that they generally initiate downstream with a gentle slope and gradually steepen as they migrate upstream. Their maximum slope is generally reached when they have propagated to the central part of the profiles (see below). Then the slope is maintained or slightly decreases during their retreat in the upper segment of the profiles.*"

[Figure]

*Figure 7. Detail retreat of an individual knickpoint from experiment BL10 (see also Fig. 6) showing its initiation with a gentle slope which subsequently steepen as it migrates upstream. Its maximum slope is reached at mid-distance between the outlet and the divide. Its lowest retreat rates are observed downstream near the outlet and upstream near the divide.*

**L16: Change these to this.**

Done.

**L20: Change rivers to river (singular).**

Done.

**L27: Remove their**

Done.

**L39-41: This is vague. It would be more insightful to the reader to explain what the limitation is and what the role of sediment supply is.**

We removed the related sentence in the revised ms.

**L46: Unclear what 'their' refers to in this sentence.**

We removed the related sentence in the revised ms.

**L48: Change has to have**

We removed the related sentence in the revised ms.

**L52: The abbreviation BL should be defined as base level before first use. Furthermore, I suggest spelling out base level throughout the manuscript, as it's not a commonly abbreviated term.**

We agree, base level is now spelling out in the new ms version.

**L73: Replace 'to create' with 'creation of'**

This sentence has been rephrased.

**L87: Define the term DEM on first usage.**

Done.

**L94: Change 'most of channels being straight' to 'because most of channels are straight'**

Done.

**L98: Change 'verified manually' to 'manually verified'**

Done.

**L98: What is meant by 'define knickpoints correctly'? Do you mean against a geometric definition?  This section is unclear because it seems there are two ways to define knickpoints.  One is based on the erosion rate relative to base level fall, and the other is a manual way (presumably based on geometry and channel slope?) that is not well defined.**

Rephrased as (Lines 100-107): "*In a second step, we computed the elevation difference between each successive pairs of longitudinal profiles and we identified knickpoints as peaks in erosion rates with values above the steady erosion amount defined by the rate of base-level fall (Fig. 2). We verified manually that this procedure defines knickpoints correctly by checking the computed positions on longitudinal profiles. We investigated in particular if the procedure is robust with respect to the time interval between successive profiles. We found that the record interval of 5 minutes is too small to produce well-defined erosional peaks, which lead us to identify knickpoint positions from a time-interval of 10 minutes.*"

**L127-128: I'm confused on what the threshold is.  Is this the minimum water depth for an area to be classified as a channel?  Can this be made more clear?**

Rephrased as (Lines 131-137): "*Given the difficulty to measure the mm-scale water depth without perturbating the flow, river widths were extracted from Floodos DEM outputs by thresholding the water depth maps, river banks corresponding to sharp variations in water depth. The water depth threshold was estimated by trial and error by comparing the rivers extracted from the calculation with direct observations on experiments where rainwater was colored by red dye (Fig. 3). A good visual agreement was obtained for a threshold value between 0.1 and 0.5 mm, and a mid-value of 0.3 mm was then used for determining river widths.*"

**L168: When using the term 'non-linear relationship' please indicate between what variables this non-linear relationship is expected.**

Rephrased as (Lines 170-171): "*Data suggest a non-linear relationship between base-level fall rate and mean retreat velocity of knickpoints*"

**L176-177: Change "whatever regardless" to "independent of"**

Changed to "regardless".

**L231: See Mackey et al. (2014) for a field case showing constant retreat (https://doi.org/10.1130/B30930.1)**

**L240: The authors write "they do not show a clear tendency of increasing" – I think the data actually looks pretty good. I would encourage the authors not to sell themselves short. It could be worth quantifying this statistically to show that there is a statistically significant increase.**

Yes, we agree that the data looks pretty good however we decided to remove this part and the dedicated figure, to lighten the ms as not essential. However, we kept the figure in the supplementary.

**L260: Replace 'channel is in average' to 'channel is on average'**

Done.

**L249-299: This paragraph is quite long and hard to follow. Can this be simplified? See my comments about writing style above. Also a discussion of the mechanism of knickpoint formation could be useful here.**

Yes, we agree and rephrased all these paragraphs. For example, lines 249-260:

"*To illustrate how knickpoints initiated near the outlet, we consider here a 120 minutes-long sequence of channel evolution in experiment MBV06 from experimental runtime 880 to 1000 minutes, during which two knickpoints successively initiate and propagate upward. Figure 10 shows 5 min intervals sequence of downstream longitudinal profiles, 40 cm-long, showing their initiation and propagation as well as the evolution of a channel cross-section located at 8 cm from the box boundary. Some photos and perspective views of the corresponding DEMs also illustrate the evolution of the channel. Complementary data are shown in Figure 11: variations over time of channel width (Fig. 11A) and unit water discharge (Fig. 11B) at the cross-section location as well as summary statistics of normalized elevation changes (Fig. 11C) and shear stress (Fig. 11D) for all pixels across the section. On the graph shown in Figure 11C, normalized values of 1 indicate erosion at the same rate than base-level fall and then steady-state conditions. Values > 1 or <1 indicate respectively higher and lower erosion rate than BL fall rate. Negative values indicate sedimentation. The sequence starts…*"

Have been changed to:

Lines 254-259: "*To illustrate how knickpoints initiated near the outlet, we consider here a 120 minutes-long sequence of channel evolution in experiment BL15 during which two knickpoints (K1 and K2) successively initiate and propagate upward (Fig. 10). In addition, we analyzed the history of channel width (Fig. 11A) and unit water discharge (Fig. 11B) at a cross-section located at 8 cm from the outlet (see location on Fig. 10B). We also present a summary of the statistics of normalized elevation changes (Fig. 11C) and shear stress (Fig. 11D) for all pixels across the section. The sequence starts…*"

Note that we also added a new section in the discussion that is specifically dedicated to the mechanism of knickpoint formation (Lines 370-413):

**"4.2 Processes controlling knickpoints initiation and propagation**

[revised manuscript text omitted]

*"*

**L265: Change 'before to subsequently widens' to 'before subsequently widening'**

Done.

**L272: Change 'erosion rates value' to 'erosion rate values'**

Done.

**L382 : I think defeat should be decrease instead?**

Yes, done.

**L383: Change 'downstream the knickpoints' to 'downstream of…'**

Done.

**L384: Change 'downstream retreating knickpoints' to 'downstream of retreating…'**

Done.

**L390-410: This feels more like results than discussion to me, and I would incorporate it in the results section.**

**L425: Change 'downstream the' to 'downstream of the'**

Done.

**L436: Change 'before to decelerated' to 'before decelerating'**

Done.

**Figures**

**Figure 3: In panel B, the solid lines are model predictions I think? This should be made clear. It would also be helpful to plot the measured value in the experiments.**

Yes solid circles in B are actually the measured values from Floodos outputs for different water depth thresholds.

**The purple line is hard to see in a panel A, why not plot both lines as red? Or choose a different high contrast color?**

We did not plot both lines as red but use different colors to refer to the two different cross-section shown in the panel A.

**Figure 6: This looks like the jet colorbar? I suggest using color schemes that are colorblind friendly (see these for example: https://www.fabiocrameri.ch/colourmaps/).**

We tried to change the colorbar accordingly unfortunately the software used to create the figure does not include such colobar.

**Figure 7: Is the colorbar the same scale in all three panels? If so, please make the labels on the colorbar for knickpoint retreat rate the same (experiment 9 has a different label than experiments 6 and 7). Also, what's the difference between the dashed blue, red and yellow lines around ndd of ~0.5 relative to the gray lines?**

Yes, the color bar scale is indeed different for the experiment BL05 (formerly designated as MBV09). As the knickpoints retreat rates for BL05 are very low compared to the other two experiments, using a same color bar scale would not have allowed to compare both knickpoint retreat rate with respect to the three experiments and knickpoint retreat rate evolution along distance to divide. We now explain in the figure legend what are the dashed color lines.

This figure is now Fig 8 in the revised ms

**Figure 8: Is panel A needed?**

**Panel A & C show essentially the same information, but panel A is very difficult to read. What about color-coding the data in panel C and eliminating panel A?**

Yes we think interesting to keep panel A in this figure (now Fig 9) because color-coding data in panel C according to A would be unreadable.

**Figure 10: The caption says blue and orange colors, but I think purple and green is meant instead?**

That is correct, figure caption was corrected.

**Figure 11: Typo in caption, it says "shear stress (FD) for all pixels", but I think this should be 'shear stress (D) for all pixels"**

True! Figure caption was corrected.

**Figure 12: Is distance on the x-axis flipped relative to fig. 6? I think it is. It would be helpful to change the x-axis label to "distance from the divide" or something else to indicate which direction the divide is in, and make sure the orientation of the x-axis is constant between figures.**

True, x-axis (Distance to divide) is now constant between all figures.

**Also in Figure 12, it would help to shade or add arrows specifying exactly where the knickpoint is. Simply putting the label K1, K2, etc on the panel is less useful than explicitly indicating where the knickpoint is.**

Here we prefer to keep the figure as it is.

**Supplementary Information**

- **Please note that throughout the supplement most of the units have powers listed as subscripts when these should be superscripts.**

  Ok, thanks.

- **At the end of the first paragraph, please spell out "Table 1" instead of "Tab. 1"**

  **Overall, the text of supplementary information has a lot of repetition from the main text (some of it is also repeated word for word). Please eliminate this redundancy.**

  Yes, we removed redundancies.

**Reply to comments by referee #2**

**RC2:** 'Comment on esurf-2021-50', **Laure Guerit**

The paper by de Lavaissière et al. presents lab experiments designed to explore the dynamics of knickpoint initiation and retreat under constant rate of base-level fall. The novelty of their experimental is the unconstrained width of rivers and this leads to new and very promising results. Based on their observations, the authors propose a conceptual model of autogenic knickpoint formation related to changes in river width. Autogenic processes are more and more recognized as major components of landscape dynamics and this work is a very nice contribution to this topic.

Yet, although I think this study has a great potential, it is a bit blurred in the current version. I identified several reasons for that which I present below together with general and specific comments. I do not see any major issue as I feel that this is mainly related to problems of syntax and general structure. Therefore, I'm confident the authors could address my comments or any from other reviews and I look forward to seeing this work published in ESurf.

All the best,

Laure Guerit

We thank the reviewer for the overall positive assessment of the presented work and for her insightful suggestions to improve the ms. We have responded to each of his detailed comments below.

**Abstract**

The use of the numerical model (Floodos) should be mentioned. I really like the use of numerical models, it is a great way to extract quantitative informations from lab experiments so you should advertise for it.

We added the following sentence in the abstract: *"To investigate the dynamics of the knickpoints, we calculated hydraulic information (water depth, river width, discharge and shear stress) using a hydrodynamic model."* We did not however mention explicitly Floodos in the abstract as it would require adding a reference here, which is not recommended.

**Introduction**

The introduction is clear and simple, which is nice. However, I think that the manuscript would gain in strength and quality with a better contextualization of the experimental approach (see also my comment below about autogenic processes). In fact, from the current manuscript, it is bit difficult to get a clear idea of the real addition of this work. This is done later in the Method but I think it should be in the introduction.

We agree. This point has also been raised by the other reviewer and as we mentioned in the response, we actually moved some sentences from the Method section to the introduction.

**In addition, the paper is mostly focused on knickpoints and I missed something about autogenic processes (beyond the conclusions of the study itself). I think the paper would be enriched by addressing this topic too in the introduction and/or in the discussion. It would broaden the impact and the potential audience. You could for example have a look at the review by Hajek and Straub in 2017 (https://doi.org/10.1146/annurev-earth-063016-015935).**

We already explain in the introduction why identifying autogenic knickpoints is important and do not feel necessary to broaden our discussion about the importance of autocyclicity in the study of Earth surface processes.

**l. 27-28: suggestion: the definition is ok but it would work for any step along a channel. Maybe add a scale ?**

We are not aware of any article that has made an inventory of knickpoint sizes in nature that could serve as a reference. But there are so many recent articles that deal with knickpoints without defining a scale that we don't think it is necessary to specify it here.

**l. 42 to 44: this sentence and the references are not really needed, maybe simply start with However or something similar. This would lighten the text a bit.**

Rephrased.

**l. 44 please change "they" to "knickpoint" (the sentence is a bit unclear with "they")**

Done.

**l. 46 I do not understand what "they" stands for**

Rephrased

**l. 50 maybe specify which forcings.**

Done. We specified "… *forcing (base-level fall and precipitation) …"*

**l. 50 I do not really understand this sentence. Do you mean that landscapes alternate between periods of steady state and periods of knickpoint propagation ? Please reformulate.**

Done. We specified "…*even when landscapes were at steady-state on average in terms of sediment flux*".

**Methods**

**I would appreciate (here, in the discussion or even in Supplement) a discussion about the choice of material and how it could affect your results.**

As mentioned above, we did not perform experiments with different materials, so we consider that adding a discussion on this topic would be suggestive. We however added the following sentence in the discussion:

*(Lines 524-527): "Simulations of Langston and Tucker (2019) highlight the role of bedrock erodibility as an important factor controlling lateral migration of rivers and the width of valleys, an issue that has not been investigated here given the similarity of the eroded materials in our experiments here."*

**I would present the three experiments here and not at the beginning of the results (therefore, Table 1 should also be moved here). This would give a better sense of what you did and it would also help with some issues with text organization (see my comments on Figures 1 for example).**

Done. We have moved the first two lines of the results section (formerly lines 155-156) to the beginning of the methods section (Lines 72-73 : "*We present here results from 3 experiments, BL05, BL10 and BL15, performed with different rates of base level fall, of respectively 5, 10 and 15 mm.h$^{-1}$ (Table 1)*").

**Finally, to ease reading, I suggest to structure the Methods into 3 subsections 1) experimental setup, 2) knickpoint extraction, 3) hydraulic data extraction.**

We did not follow this recommendation as it would make three relatively small paragraphs.

**l. 70 please give values for the base level fall rate.**

Done. Note that we have renamed the experiments by indicating the rate of base level fall in their name.

**l. 73 the description of the initial configuration should come before the experiment itself.**

Done.

**l. 98 what does "correctly" means ? This sentence suggest that you define knickpoints based on erosion rate and that you consider them as "correct" based on another criteria (slope I guess) ? The procedure is not that clear, please consider rephrasing.**

Done. Changed to "*we computed the elevation difference between each successive pairs of longitudinal profiles and we identified knickpoints as peaks in erosion rates with values above the steady erosion amount defined by the rate of base-level fall (Fig. 2). We verified manually that this procedure defines knickpoints correctly by checking the computed positions on longitudinal profiles. ...*"

**l. 111 to 126: almost word by word repetition with the supplement material.**

True! We have removed these lines in the supplement

**l. 116: the outputs are repeated line 118. Consider rephrasing**

True! line 116 has been removed

**l. 117 the use of "spatial distribution of precipitation" is strange as as you state on the previous page that precipitation are constant. Do you refer to heterogeneities (inherent to the experimental design) mentioned in the supplement ? This should be clarified.**

We add in the method section (Lines 77-78) that we used a *"mean rainfall rate of 95 mm.h$^{-1}$ with a spatial coefficient of variation (standard deviation/mean) of 35%"*.

**l. 111 to 136: I found this while paragraph difficult to follow. Please try to be more straightforward or to rephrase a bit.**

We rephrased this paragraph and hope it is clearer now:
Lines 117-137: *"DEMs were also used to compute hydraulic information (water depth, river width, discharge and shear stress) using the Floodos hydrodynamic model of Davy et al. (2017; see also Baynes et al. (2018, 2020) for previous use of Floodos for analyzing laboratory experiments). Floodos is a precipiton-based model that calculates the 2D shallow water equations (SWE) without inertia terms, from the routing of elementary water volumes on top of topography. We ran Floodos on successive DEMs of experiments by considering spatial distribution of precipitation, then generating several output raster products at the pixel size, including water depth, unit discharge and bed shear stress that were then used for computation of hydrologic parameters (river width, specific discharge and shear stress). The solution of the SWE depends on the friction coefficient (C) that depends on water viscosity only for laminar flow; its theoretical value is ~2.5 x 10$^6$ m$^{-1}$.s$^{-1}$ at 10°C (Baynes et al., 2018). To ensure that Floodos outputs (e.g. water depth raster maps) calculated using this value are consistent with actual experiment hydraulic conditions, we injected dye in the rainfall water during a run to catch the actual extent of water flow and make rivers visible. A visual comparison with Floodos results shows a good match between model outputs and experimental results (Fig. S2), which validates the numerical method and the expected theoretical friction coefficient C (Baynes et al., 2018). Given the difficulty to measure the mm-scale water depth without perturbating the flow, river widths were extracted from Floodos DEM outputs by thresholding the water depth maps, river banks corresponding to sharp variations in water depth. The water depth threshold was estimated by trial and error by comparing the rivers extracted from the calculation with direct observations on experiments where rainwater was colored by red dye (Fig. 3). A good visual agreement was obtained for a threshold value between 0.1 and 0.5 mm, and a mid-value of 0.3 mm was then used for determining river widths."*

**Results**

**The text misses a clear organization and suffers from repetitions. This prevents me from getting a clear sense of the main results of the study, although I believe the material is already here (I think that lines 195 and 295-296 are the main results of this study).**

We agree that the text was not sufficiently well organized with repetition errors. We have thoroughly reworked the presentation of the results accordingly.

**Some ideas to help rephrasing:**

**- figures are sometimes described instead of being used to support a result (l. 159-161, l. 190-191, l. 251-257) and some of them are barely used (Figure 6 for example).**

We have edited these sentences accordingly, e.g. *"The successive longitudinal profiles of the main river investigated in each experiment (Fig. 6) illustrate the growth..."* instead of *"Figure 6 shows the evolution of the longitudinal profile of the main river investigated in each experiment, .... These stacks illustrate the growth ..."*

**- repetitions (for ex. l. 188 and l. 195, l. 197-198 and 198-199)**

Updated.

**- use numbers to quantify your statements. There are only three experiments, it is really easy to be more quantitative. For example, line 159, 166: give the three values. l. 167 "increase in average": there is only three points so I assume you can be more specific, l. 176 "very comparable" is quite vague.**

l. 167: updated (removed "in average").

l. 176 "very comparable", we explain in the following sentence what is meant by "very comparable"

**- the initiation of knickpoint section could come before the section on the dynamics (knickpoints must form before they can propagate)**

Updated. Lines 159-166. "*Figure 6 shows the evolution of the longitudinal profile of the main river investigated in each experiment, as well as topography of the initial surface, the profiles being colored according to the experimental runtime. These stacks illustrate the growth of rivers as they propagate within the box. Longitudinal profiles show alternations of segments with low and higher slopes, the later defining knickpoints that propagate upward, some profiles showing even several knickpoints that retreat simultaneously. Knickpoints regularly initiate at the outlet throughout the duration of the runs in all experiments, and propagate upward until they reach and merge with the divide.*"

rephrased:

Lines 159-164 "*The successive longitudinal profiles of the main river investigated in each experiment (Fig. 6) illustrate the growth of rivers as they propagate within the box. These profiles show alternations of segments with low and higher slopes, the later defining knickpoints. They regularly initiate at the outlet throughout the duration of the runs in all experiments and propagate upward until they reach and merge with the divide, some profiles showing even several knickpoints that retreat simultaneously (Fig. 6).*"

**- split the text when a new idea is developed (some suggestions : l. 171, l. 277, l. 284)**

Done.

**The normalized distance to divide (l. 174) and the normalized longitudinal distance of maximum velocity of knickpoint (l. 237 - maybe find a simpler expression ?) are important parameters for your study. I think they should be defined in the Methods, not in the Results. Once again, this is give a better organization to the manuscript and ease the reading.**

We agree on the principle but concretely, to define the normalized distance nDD, we need to have descriptive elements of the experiments which are not yet introduced in the methods. We therefore prefer to leave the definition of this parameter here.

We have not found a better expression than "normalized longitudinal distance of maximum velocity of knickpoint". The point of this expression is that it indicates very precisely what we are referring to.

**l. 299-233 seem to be rather discussion**

Agreed. Moved to discussion section.

**l. 233 which effect ? please specify**

This has been removed in the new ms.

**l. 240-243 this paragraph is a bit confusing. The authors state that there is no clear tendency but that there is positive correlation (that I could indeed observe in both graphs). Please reformulate, make your point clearer and do not minimize your results.**

This has been removed in the new ms.

**l. 264 and 269 "first knickpoint" should be "second knickpoint" as you mentioned that a knickpoint is already there l. 261.**

The "first knickpoint" that we referred to is the first in the sequence shown in the Figures 10 to 12 and discussed in the text. Actually, many knickpoints have already retreated before this sequence which started at runtime 880' and the remnants of one is actually visible in the upstream part of the profile shown at 880'. Then, the first knickpoint discussed in sequence has initiated at 895'. To be clearer, we now refer to this first knickpoint of the sequence as K1, and the second as K2:

Lines 254-261: "*To illustrate how knickpoints initiated near the outlet, we consider here a 120 minutes-long sequence of channel evolution in experiment BL15 during which two knickpoints (K1 and K2) successively initiate and propagate upward (Fig. 10) ... The sequence starts with a "standard" profile (i.e., a typical river profile without any perturbation) at runtimes 880 and 890 min once a previous knickpoint already propagated, still visible upstream in Figure 10A*"

**l. 265 quantify "widens"**

Done.

**l. 267-271 syntax is a bit complex, please consider rephrasing for simplification**

"*The narrowing phase goes with an increase in unit discharge (Fig. 11B) and with an enhanced erosion well above the BL fall rate, ...*"

Rephrased to:

Lines 266-268: "*The narrowing phase is naturally associated with an increase of the unit discharge (Fig. 11B) and with an enhanced erosion well above the base level fall rate, ....*"

**l. 276 and 277 third ?**

K2, see above. Updated in the text.

**l. 283 the mean is positive but the median is negative. What does it imply, could you comment on that ?**

This is a detail. Most of the pixels in the section are in sedimentation (negative value of median) but on average the section is still in erosion (positive mean). We do not think it is necessary to develop this point further.

**l. 288-299 sounds a bit like discussion. Please avoid repetition (l.288-299 and 295).**

Yes, but we think it is necessary to make this point here immediately after the presentation of data because it is quite complex and we think it is necessary to put it forward so that the reader understands

where this important finding comes from. This point is indeed taken up again later in detail in the discussion.

Line 295. It is indeed a bit of a repetition, but we think it is necessary to remind here what the steady state means (erosion rate is equal to base level rate) for the readers who are not familiar with these concepts.

**l. 436-440 almost word by word repetition of l. 229-231.**

Yes indeed. Lines 229-231 have been removed in the new ms (see above).

**Discussion**

**Here again, I would suggest to structure the discussion into subsections to ease the reading. A general comment for this last section is that the authors do not highlight their specific contribution. I really think this study has a great potential to document in a quantitative way the initiation and dynamics of autogenic knickpoints and the authors even provide a mechanism (via river width) for that. However, this is somehow hidden in the text.**

We thank the reviewer for her warm encouragement and suggestions to better promote our results and we have also reworked the discussion in depth by reorganizing it into three distinct parts:

**4.1 Autogenic knickpoints**

**4.2 Processes controlling knickpoints initiation and propagation**

**4.3 Implications**

Note that we also expanded the discussion by addressing important points in more detail:

What are the causes of widening downstream of knickpoints? What are the respective roles of narrowing and steepening in knickpoint initiation?

**The paper could do better here with the idea of autogenic knickpoints: what are the implications for stratigraphy or for tectonic studies ? How to recognize autogenic from allogenic knickpoints ? I really think the paper would gain in strength by addressing these questions in the discussion.**

This is a point that Reviewer#1 also made and as mentioned earlier, we have added a section at the end of the discussion (4.3 Implications) that specifically addresses these issues. Our work is still too preliminary to be able to define criteria for discriminating between autogenic and allogenic knickpoints, but we have developed a few suggestions that would be interesting to explore.

Lines 478-483: "*Finding criteria that could be used in the analysis of natural systems to differentiate these autocyclic knickpoints from those formed in response to tectonics or climate would be an important step in the continuation of this work. A specificity of knickpoints in our experiments is to initiate downstream with a gentle slope, which then amplifies in the early stages of migration, and as a hypothesis we suggest that this may be characteristic of their autogenic formation following the mechanism described here.*"

**The concluding paragraph does not give credits to your work and I can not get a sense of your results from these lines. "Our model is consistent with this proposition" is quite a limited conclusion for your study.**

We added a dedicated conclusion section in the revised ms.

**l. 369 why admitting ? you just demonstrated this !**

Thank you, this was corrected ( "Admitting that…" changed to "Given that …")

**l. 393-410 to me, it is a major outcome of this study, that could almost belong to the Results. This should be highlighted, maybe with a conceptual figure ? As mentioned elsewhere, I think that figures could be simplified and some even removed so it's ok to consider a new one.**

We agree that this is a major point, which follows from the data presented in the results section however we prefer to keep these sentences in the discussion section. The other reviewer also suggested that a summary figure be added, but as we have explained above we have not managed to create a figure that satisfies us to explain all the details of the mechanism of knickpoints generation.

**I may have miss it but why do rivers widen at all ?Also, you mention that erosion is inhibited after the passage of a of knickpoint but widening implies lateral erosion. Maybe specify vertical erosion or discuss this point a bit further.**

This is indeed a very important issue that we barely discuss in the previous version. Much of the new section 4.2 of the discussion now focuses specifically on this issue:

Lines 428-453: "*Further work is required to understand the mechanisms responsible for lateral channel erosion in our experiments, which is a key ingredient for understanding river mobility and widening. Several field (e.g. Hartshorn et al., 2002; Turowski et al., 2008; Fuller et al., 2009), experimental (e.g. Wickert et al., 2013; Bufe et al., 2016; Fuller et al., 2016; Baynes et al., 2020) and numerical (e.g. Turowski et al., 2007; Lague, 2010; Langston and Tucker, 2018; Li et al., 2021) studies have demonstrated that high sediment flux relative to transport capacity promotes increased lateral channel erosion. Most of these studies highlight the role of cover effect, the protection of the river bed by transient deposition of sediments on the river bed (Sklar and Dietrich, 2001; Turowski et al., 2007, 2008; Lague, 2010; Baynes et al., 2020; Li et al., 2021), as a main factor promoting lateral erosion in high sediment flux settings. Other studies show that by modifying the bed roughness, sediment deposition may deflect the flow, which also promotes lateral erosion and widening (Finnegan et al., 2007; Fuller et al., 2016). Contrary to experimental devices specifically designed to address these issues, large flumes in particular (e.g. Finnegan et al., 2007; Fuller et al., 2016), direct observation on actual processes that drive lateral erosion in our experiments is made difficult by the small size of the topographic features, the depth of rivers being of millimeter scale, and by the low grain size of the material used. Opacity due to the generation of the artificial rainfall also considerably limits direct observation during the runs. Despite these limitations, data suggest that lateral erosion and river widening in our experiments is also related to increase in sediment flux. We show actually that knickpoints are location of enhanced erosion well above the rate of base level fall. We document for example mean erosion rates greater than 5 times the base level fall rate, with extreme values up to a factor of 8 locally (Fig. 11 and 12). Downstream, where rivers widen, we observe that the general decrease in erosion rate is also associated with local deposition in some parts of the channels (for example at runtime 915 min in Figure 11 or 975 min in Figures 10 to 12). We then hypothesize that lateral erosion and widening are due in part to the increase sediment flux related to enhanced erosion on knickpoints. Further work is needed to test this hypothesis, for example by investigating in detail spatio-temporal variations in erosion and sedimentation during width widening.*"

**l. 440-445 this is already said in the results. Consider removing these sentences.**

Done.

**l. 425-426: this is already said l. 421-422 and said again l. 433-434. Please simplify.**

Right, we have removed lines 425-426 and 430-434 that were not necessary.

**Additional comments**

**I noticed some minor language mistake. It did not alter my reading or my understanding of the work but it should be corrected. The text could also be simplified with a bit of rephrasing. Syntax is sometimes complicated, I noted a lot of repetitions and you could change structures like "On the graph shown in Figure 11C (l. 257), […]" to "normalized values of […] (Fig. 11C) . This would lighten the text and give some space to develop the suggestions proposed by the reviews.**

**Nomenclature: I guess the names of the experiments are related to your work flow but it does really served the manuscript. To make it easier to read, what about experiments 1, 2, 3 (or 5- 10-15 to refer to the rate of base level fall) ?**

We agree, in the new ms version we changed the internal experiments names, MB06, MB07 and MBV09 respectively to BL15, BL10 and BL05 (to refer to the rate of base level fall).

**l. 32 sel-level -> sea level**

Done.

**l. 40 limitation -> limitations (?)**

Text was changed here.

**l. 74 remove "here"**

Text was changed.

**l. 116 missing capital f (floodos)**

Modified.

**l. 171 (and elsewhere in the text): "In order to be able to compare" can be reduced to "to compare". Such a simplification would ease the reading and shorten a bit the manuscript.**

We agree, we modified the text accordingly.

**l. 174 Figure 2 should be Figure 4**

Done.

**l. 176 remove regardless**

Right, the text is now modified from "within an experiment whatever regardless the stage" to "within an experiment regardless the stage" (former L176-L177 ; now L180).

**l. 329: missing word after "once a knickpoint has" (I guess it is "passed")**

True, we changed the line to "once a knickpoint has propagated" (now L332)

**l. 413: "is" instead of "in"**

Done.

**l. 417: missing space (Fig.12)**

Done.

**Figures and table**

**As a general comment, the figures are nice and clearly illustrate the experiments and results, but they are sometimes a bit dense and could benefit from simplifications and harmonization (see specific comment below). The size of the text changes from one figure to another and is sometimes very, very small. I suggest considering a fixe, readable size and to adjust the content of the figures.**

Done, we have harmonized the fonts as much as possible.

**Finally, some figures are not exploited at their full potential. For example, Figure 1 is only referred to once. The individual panels of Figure 1 and 3 are not referred to and there is no reference to Figure 13. The text should take better advantage of the illustrations and/or figures could be simplified or moved to the Supplementary.**

Figure 9 has been moved to the supplementary and we added a new figure (Fig. 7, see above).

*Figure 1*

**Panel A: please indicate the total height of the box.**

Done.

**Panel C: I can see numbers of the left and right sides of the box, what are they for ?**

It is an internal code used to identify the different components of the device (it is printed on the components of the device). We don't think useful to explain this here.

**caption: the experiments are not introduced at the point of the text. Please adjust either the text of the caption.**

We agree, in the new version of the manuscript, the experiments are introduced above.

*Figure 2*

**Text is really small.**

Changed.

**caption: the experiments are not introduced at the point of the text. Please adjust either the text of the caption.**

We agree, the text was modified accordingly. See the answer above.

*Figure 3*

**Panel A: the superposition of the DEM colorsmap together with the water depth colormap makes the figure a bit complicated to read.The elevation does not seem to be used so I suggest to remove the colors. Maybe use grey shade/ slope illumination only ? If you keep the colors, please add the color scale.**

That is true. The color scale of the DEM is now added to the figure.

**Panel B: axis numbers are very small.**

Changed.

**caption: it is very long ! Most of it should be in the main text.**

The caption is very long, indeed, but we think all the information are needed to a better understanding of this figure. We prefer to let all the caption here to lighten the main text.

**l. 146: shallow instead of thin ?**

Done.

*Table 1*

**nDDmax is not a straightforward variable. Please use the full name or define it in the caption**

See our response to comment above. $nDD_{Vmax}$ may be a bit complex but it has the merit of stating precisely what it measures.

**Mean knickpoints retreat rate: do these values come from Fig. 8 ? If so, maybe use similar units to ease the reading.**

Yes, the values come from Figure 9 (Figure 8 in the initial ms).

**Also, 4 digits after the comma might be a bit excessive.**

That is right, values were formatted accordingly.

*Figure 5*

**caption: please describe the inset figure**

Done.

**Figure 7**

**The colors scale of experiment 9 should be the same as experiments 6 and 7 to ease the comparison.**

Now Fig. 8

The color bar scale is indeed different for the experiment BL05 (formerly designated as MBV09). As knickpoints retreat rates for BL05 are very low compared to the other two experiments, using a same color bar scale would not have allowed to compare both knickpoint retreat rate with respect to the three experiments and knickpoint retreat rate evolution along distance to divide for a same experiment.

**caption: please indicate what are the blue, orange and yellow lines for.**

Done. We added in caption "*Thin colored dashed lines show nDDVmax, the normalized distance where the highest rate of retreat velocity is deduced from the analysis (see text and Figure 9C)*"

**Figure 8**

**Text is quite small.**

That is true. Labels were modified accordingly

**The 3 panels are more or less similar and are just different ways of presenting the data. In order to simplify and gain space, I suggest to choose one way and to remove the other ones (or to move them to the supplementary).**

This is very important data, which is actually represented here in three different ways. Given the large amount of information, it was not possible to find a way of representing the data that would show everything clearly. We aim to keep the data represented in these three ways.

**Figure 9**

**caption is a bit unclear, please use full wording instead of symbols.**

We agree, we changed the caption accordingly. Note that this figure is now the Figure S4 in the supplementary.

**Figure 10**

**Panel B is referred to before Panel A, please adjust the text or the figure.**

Yes, but we referred to Fig. 10 before referring to 10B then after.

**Panel A: what do the changes in colors correspond to?**

The change in colors correspond to distinct knickpoints upstream propagation. That was a bit unclear in the previous ms version The caption is now corrected to:

"*Propagation of the first (K1; initiated at 895') and second (K2; initiated at 935') knickpoints is shown in green and purple colors respectively.*"

**Panel C: missing color scale for the elevation. The indication of cross section is a bit difficult to read, maybe try in white ?**

Done.

**The caption mentions the propagation of two knickpoints together with blue and orange colors that I do not see. Please correct.**

The caption was corrected accordingly. See the response for the Panel A comment.

*Figure 11*

**Text is small**

That is true. The text was modified accordingly.

**Panel C: y-axis is "normalized erosion or sedimentation" but in the caption, it is given as "normalized elevation changes". please correct.**

Done. The caption was corrected to "plots of normalized erosion or sedimentation".

**caption: FD should be D. "Orange solid […] percentiles": as in Figure 8B, this could be in the legend of the panels.**

Done.

*Figure 13*

**I found no reference to this figure in the text.**

It is because this figure is only referred as "Figure 13A" or "Figure13B"

---

## Referee Report (RR1)

**Generation of autogenic knickpoints in laboratory landscape experiments evolving under constant forcing.**

**Léopold de Lavaissière[1], Stéphane Bonnet[1], Anne Guyez[1], and Philippe Davy[2]**

[1] *GET, Université de Toulouse, CNRS, IRD, UPS(Toulouse), France,*

[2] *Univ Rennes, CNRS, Géosciences Rennes - UMR 6118, 35000 Rennes, France,*

Correspondance to Léopold de Lavaissière (leopold.delavaissiere@gmail.com)

**ABSTRACT**

The upward propagation of knickpoints in river longitudinal profiles of rivers is commonly related to discrete changes in tectonics, climate or base-level. However, the recognition that some knickpoints may form autogenically, independently of any external perturbation, may challenge these interpretations. We investigate here the genesis and dynamics of such autogenic knickpoints in laboratory experiments at the drainage basin scale, where landscape evolved in response to constant rates of base-level fall and precipitation. Despite this constant forcing, we observe that knickpoints regularly initiate in rivers at the catchments' outlet throughout experiments duration. The upstream propagation rate of knickpoint does not decrease monotonically in relationship with the decrease of their drainage area, as predicted by stream-power based models, but it first increases until the mid-part of catchments before decreasing. To investigate the dynamics of the knickpoints, we calculated hydraulic information (water depth, river width, discharge and shear stress) using a hydrodynamic model. We show that their initiation at the outlet coincides with a fairly abrupt river narrowing entailing an increase in their shear stress. Then, once knickpoints have propagated upward, rivers widen entailing a decrease in shear stress and incision rate, making the river incision lower than the base-level fall rate. This creates an unstable situation which drives the formation of a new knickpoint. The experiments suggest a new cyclic and

*→ you could use the term 'autocyclic' instead*

**autogenic model of knickpoints generation controlled by river width dynamics**  *independent* **of**

**variations**  *in* **climate or tectonics** **. This questions an interpretation of landscape**

**records focusing only on climate and tectonic changes without considering autogenic processes.**

*→ it could be worth mentioning lithology in the abstract because there's been a lot of work related to lithologic control on KP formation.*

[revised manuscript text omitted]
 [→ *doesn't narrowing cause an ↑ in unit discharge by definition? If you are suggesting a positive feedback between narrowing + an increase in unit discharge, please expand + be explicit.*] (Fig. 11B) and with  enhanced erosion  *greater than* the base level fall rate, up to 4 times  *the base level fall rate* rate in average at 900 min (Fig. 11 C), with extremes as high as 8 times the base level *fall* rate. Once  K1 has retreated, unit discharge decreases as the channel subsequently widens, to reach a width of 25 cm to 28 cm between 925 and 930 min (Fig. 11A) while a new regular profile, i.e. without any slope break, *i*^established at 930 min (Fig. 10A). The normalized erosion across the section decreases below the base level value (Fig. 11C), with mean erosion rate values of 0.53, 0.36 and 0.76 times below the base level rates between 915 to 925 min. Longitudinally, the profiles stack together downstream of the knickpoint following its retreat from 895 to 925 min (Fig. 10A), which also indicates minor vertical erosion here once the knickpoint has retreated despite the ongoing base level fall. The second knickpoint (K2) then initiates at 935 min, propagates upstream in a similar way, and disappears leading to the setting up of a new regular profile at 980 min (Fig. 10A). Channel narrowing is also observed on the cross-section at the passage of this second knickpoint with a width that decreases to ~15 mm wide (Fig. 10B and 11A), associated with an

*I find the language here confusing because K2 is still clearly visible at 980 min starting at position x = 325 mm. Does this text only refer to the KP at the cross-section location (x=75 mm)? If so, I don't think K2 has disappeared, rather it has simply propagated upstream. Please clarify this text.*

increase of the unit discharge and the erosion rate (Fig. 11C). It is followed again by a phase of widening to reach a width to around 30 / 35 mm once the knickpoint has propagated upstream and by a decreasing erosion below the base level fall rate (Fig. 11C). Again, the longitudinal profiles stack together downstream of the knickpoint (Fig. 10A). Note that at 975 min, most of the surveyed section is undergoing sedimentation (mean normalized erosion rate is 0.1 and median is -0.25: Figures 10B and

11C).  The distribution of river bed shear stress along the section is given in the Figure 11D. Despite a large variability along the section, one can observe a significant increase of the median and maximum values at the time of the knickpoint passage, both for K1 and K2. Once knickpoints passed, the shear stresses decrease as the river widens.

This sequence illustrates that the rivers are never in equilibrium at the 5 min time-scale, but continuously oscillate over time between disequilibrium states with periods when channel are too wide to keep pace with the base level, and periods of knickpoint propagation when the erosion is enhanced to catch up the base level. The river width is the regulation parameter which allows the river erosion to adapt by increasing or decreasing the unit discharge. These knickpoints then propagate upward up to the divide as discussed previously (Fig. 6). The average erosion rate is similar to the baselevel fall rate (0.99) but it does not correspond to any stable configuration of the river since the erosion rate fluctuates between smaller and larger values. Knickpoints are by-products of this unsteady dynamics, which are generated during the phases when the river catches up with its erosion deficit with respect to the base level.

[Figure]

**Figure 10.** *Downstream knickpoints initiation and propagation in a* 120 minutes*-long sequence of*

experiment BL15 from experimental runtime 880 to 1000 minutes. *(A) Sequence of downstream*

*longitudinal profiles (5 min time-interval) of the investigated river, corresponding to the sequence*

*hydro-geomorphic parameters shown in Figures 11 and 12. Propagation of the first (K1; initiated at*

*895') and second (K2; initiated at 935') knickpoints is shown in green and purple colors respectively*

*(see text). (B) Time evolution of successive cross-sections of the channel at 80 mm from the outlet. (C)*

*Photos and perspective views of DEM at five time-steps.*

[Figure]

*It's not clear from the methods how this timescale of the KP passing a single xs is defined.*

*it would be more intuitive (in my opinion) to simplify this as: 10⁶ mm².h⁻¹*

*this panel has two yellow lines, one for the median values and a second connecting these median values. I find this confusing & I suggest removing the second line.*

*you might consider plotting water velocity instead of or in addition to shear stress as it has been shown to be a better predictor of sediment transport than shear stress. See Yager et al 2018 in Geomorphology "The trouble with Shear stress"*

**Figure 11.** *Time-series (5 min time interval) of river width (A) and unit and total discharge (B) for the*

*channel in experiment BL15 shown in Figure 10B. Time-series of box-and-whisker plots of normalized*

*erosion or sedimentation (C) and shear stress (D) for all pixels across the section. Orange solid circles* 〔cross section (?)〕

*and yellow lines show the mean and median values respectively. Edges of the boxes indicate the 25th*

*and 75ᵗʰ percentiles. Note that in C, normalized values of 1 indicate erosion at the same rate than base-* 〔as〕

*level fall  (steady-state conditions) .Values > 1 or <1 indicate respectively higher and lower*

*erosion rate than BL fall rate. Negative values indicate sedimentation. On all graphs, crosshatched*

*areas indicate the passage of knickpoints K1 and K2.*

To complement cross-section data, we also illustrate (Fig. 12) how parameters vary longitudinally by considering four stages, two before (925 min) and after (975 min) the passage of the knickpoint K2 and two during its retreat (945 and 950 min). Note that at 925 min, the previous knickpoint (K1) has just passed upstream the investigated profile and is responsible for the enhanced normalized erosion and increased shear stress upstream between distance 200 to 350 mm. Similarly, at 975 min the second knickpoint (K2) is still in the upstream part of the profile between distance 300 to 350 mm. We also reported the longitudinal variations in river width, shear stress and normalized erosion along the profiles (Fig. 12). At runtimes 925 and 975 min, before and after the passage of knickpoint K2, erosion is below the base level rate along all the profiles down the knickpoints, with even localized sedimentation at 975

min between 50 and ~150 mm. These sections are characterized by low shear stress values, being between 0.5 and 1 and by rivers that widen downward (around 0.7 mm/cm). On the opposite, during the passage of knickpoint K2, at runtimes 945 and 950 min, mean shear stress increases locally at the knickpoint location, being > 1 and the normalized erosion overpasses the base level rate there. These knickpoint segments are characterized by a narrowing of the rivers as already shown previously. The data illustrate that erosion mainly occurs during periods of knickpoint retreat though a combination of local steepening of the profile and narrowing of the river, resulting in an increased shear stress. On the opposite, once a knickpoint has propagated and between the passage of two successive knickpoints, erosion decreases significantly and does not longer compensate the base level fall. These periods of defeated erosion are characterized by low bed shear stress values in wide rivers, that widen downward.

[Figure]

*I suggest adding arrows to point to the KP on the profile.*

*Steady state (erosion = base-level fall)*

Erosion < Base-level fall

Sedimentation

[revised manuscript text omitted]

define the variable $\emptyset$ diameter?

[Figure]

[Figure]

[Figure]

**Figure S1.** *Overviews of the experimental setup.*

[Figure]

**Figure S2.** *Floodos hydrodynamic model water depth output for three different friction coefficients C applied on the same DEM of an experiment. Black lines indicate the actual channel boundaries observed during the corresponding experimental run by injected red dye in the water used to produce the artificial rainfall (right). Channels visible on water depth maps tend to have a good match with actual observed channels when using the theoretical value of the fiction coefficient (2.5 x 10⁶ m⁻¹ s⁻¹).*

[Figure]

**Figure S3.** *Extraction of rivers longitudinal profiles (bottom), showing the propagation of an individual knickpoint (the one highlighted in the Figure 7, from the experiment BL10). The two photos illustrate the evolution of the knickpoint shape through time (grey gradient) and according to its position along the distance from the outlet.*

[Figure]

*please define these variables in the caption.*

**Figure S4.** *Relationship between knickpoints retreat rates and unit discharge (total discharge normalized to river width) for nDD < nDD$_{Vmax}$ (left) and nDD > nDD$_{Vmax}$ (right). Data for knickpoints above nDD$_{Vmax}$ allows to consider retreat rates against more than two orders of magnitude of unit discharge and are consistent with an increasing rate of retreat with discharge. Data below nDD$_{Vmax}$ show 3 distinct fields without any clear trend with discharge. The restricted range of discharge data however limits the analysis.*

*shows*

---

## Author Response (AR2)

**Associate Editor decision: Publish subject to minor revisions (review by editor)**
by Fiona Clubb
**Comments to the author**:

**Dear authors,**
**I have now received your revised manuscript, response to reviewer's comments, and re-review comments from Reviewer 1. I would like to thank you for engaging constructively with the reviewers. The new manuscript is significantly improved from the initial version, with the novelty of the experimental approach and the importance of the topic for our understanding of landscape evolution clear. I think this paper will be of great interest to the geomorphology community and is a fitting contribution to ESurf. I am happy to accept your manuscript for publication in ESurf following some minor corrections to address which are detailed below.**

**Best wishes,**
**Fiona**

Thanks for your kind comments.
We explain below the latest corrections we have made in response to your comments and re-review by reviewer 1.

**General comments**

1) Both reviewers noted issues with style and language. Some of these have been addressed through the revision process, but there are still some issues throughout which make it hard to understand some of the text. Please carefully read through your manuscript and do a check for clarity. I have made some specific suggestions below and suggest that you address the handwritten comments from Reviewer 1's re-review (although a line-by-line response to these is not needed when submitting your revision).

We hope that the new corrections will have improved these issues.

2) Reviewer 1 raised some good points about the retreat rate patterns that you observe during the experiments. I agree with the reviewer that the pattern of knickpoint retreat acceleration followed by deceleration are clear in Experiment BL15, but not clear from the other experiments. In their initial review the reviewer asked for more robust statistical testing, which was not addressed in the revised manuscript. Please address this comment by testing the significance of the modelled quadratic fit to the observed knickpoint retreat rates for each experiment
.
Done. We modified the related figure (Fig. 9) to better illustrate the acceleration and deceleration trends in experiments BL05 and BL10 (see also comments below). We have also added the statistical tests suggested by reviewer 1 in a new supplementary figure (now Fig. S4 in the Supplemental Material).

**Specific comments:**

Figure 1C: please annotate this photo with the position of the sliding gate at the outlet, to make it clear to the reader which orientation the box is in in this photograph.

Done.

Extraction of river long profiles from the experiments: only one river profile is extracted from each DEM, but from the photos in Fig 1 it looks like there are multiple catchments within each experimental setup. It's not clear how the "main river" was identified from the experiments. Please clarify in the methodology.

Done. We considered one river per experiment, generally the one with the largest catchment.

Figure 6: please avoid the use of the Jet colourmap. I suggest using one which is more perceptually uniform (e.g. viridis, plasma, etc.).

Done, we now use the Plasma colormap in Figure 6.

**Technical corrections:**

Line 21: typo "heir"

This was corrected

Line 66: should be "evolve" rather than "evolved"

This was corrected

Line 89: missing space between Table and the number

This was corrected

Throughout the text there are a few times when an extra "s" is added to "knickpoint". E.g. Table 1, Fig 10 caption. Please check for grammar.

This was corrected

Line 133: Fix sentence structure.

This was corrected

Figure 3 caption is very long and quite confusing: please condense and rephrase.

This was corrected. We have shortened the caption

Line 161: should be "latter" rather than "later"

This was corrected

Figure 7 Caption: Remove the word "Detail".

This was corrected

**Reviewer #1**.

These are my second round of comments on the submitted manuscript by Lavaissière and colleagues. As I stated in my original review, I am supportive of the publication of this manuscript in ESurf as I think the authors have performed interesting and innovative experiments and analyses that will improve the community's understanding of how knickpoints form and retreat, and what influence this may have on landscape evolution. I also believe the manuscript has improved significantly since its original submission.

**In this revised version, the associated editor asked me to specifically discuss 1) the additional discussion about the mechanisms of knickpoint initiation and retreat, and 2) the testing whether a quadratic function is an appropriate fit to the retreat rate curves,** and, both of which I commented on in my original review.

**Regarding point number 1** above, I believe the authors have (more or less) suitably addressed my comments, and the additional details on the mechanism of knickpoint formation and retreat is appropriate. I think there is still some minor work to do here. I found Section 4.2 where the authors explain many of these details to be a bit hard to follow. In particular, the authors argue in Section 4.2 that 'rivers no longer incised' (L381) after the passage of knickpoints. Figure 11C does a nice job of showing that this is not true. While there are a few cases after the passage of a knickpoint where the river alluviates, the more common trend is for erosion rates to drop below the rate of base level fall, thereby allowing the profile to steepen so that a new knickpoint can be made. I would encourage the authors to change the wording throughout the section to highlight the importance of erosion rate falling below the rate of base level fall (as opposed to erosion rates dropping to zero), as I believe this is the primary mechanistic control.

→ Done. The related sentence in section 4.2 :

*"immediately after the retreat of a knickpoint, we show that erosion is inhibited downstream and **rivers no longer incised** despite the ongoing base level fall, until the passage of a new knickpoint."*

was modified to

*"Immediately after the retreat of a knickpoint, we show that erosion in the section of the channel where the knickpoint just passed is inhibited despite the ongoing base level fall: river incision is lower than the rate of base level fall, until the passage of a new knickpoint"*

A second point the authors can make here is that it appears the majority of the erosion in the experiments is made by the upstream propagation of autogenic knickpoints, highlighting how important it may be to understand the dynamics of these systems to predict landscape evolution.

This is indeed an important point that we put forward in the conclusion:

*"Rivers in our experiments thus evolve following sequences of width widening and narrowing that drive the initiation and propagation of successive knickpoints. As a result, incision is fundamentally discontinuous over time despite continuous forcing. It occurs during discrete events of knickpoint propagation that allow the rivers to recover from the incision delay accumulated during widening periods."*

Related, in Section 4.3, the authors seem to contradict themselves. The authors suggest that the bell-shaped curve of knickpoint retreat vs distance (Fig. 9) may be a diagnostic characteristic of autogenic knickpoint creation. However, the authors then go on to argue that their mechanism is analogous to a system with discrete pulses of base level fall (L503-507). I agree, but if that's true, then we would also expect discrete cases of base level fall to produce the bell-shaped curve of knickpoint retreat vs. distance, which I don't think has been previously observed. To me this suggests that there's something else going on that is unique to the experiments here that may be creating this interesting retreat pattern. If the authors can't fully explain why this retreat pattern emerges, I think that's OK and the paper should still be published, but I would encourage the authors to try to revise the text to avoid this contradiction.

We disagree with this comment. We explain in the ms that despite continuous base level fall in our experiments, the geometry of successive longitudinal profiles is similar to the geometry observed when a geomorphic system is forced by discrete drops of base level fall, (section 4.3):

*"… Thus, the sequential evolution of longitudinal profiles is very similar to the geometry that would be observed if the system was forced by discrete drops of the base level, rather than by a continuous drop as it is actually the case."*

but this similarity of geometries does not imply that the dynamics of knickpoint migration is similar in both cases as the reviewer states. We disagree with the reviewer when he/she writes: "*then we would also expect discrete cases of base level fall to produce the bell-shaped curve of knickpoint retreat vs. distance*" and we have never written nor suggested that in the ms.

**Regarding point number 2,** I do not think the authors response and subsequent changes to the manuscript are sufficient. The results from Experiment BL15 are fairly clear that knickpoint retreat rates speed up and then slow down. I do not think this is as obvious in Experiments BL10 and BL05. The authors suggested that their hypothesis of knickpoints speeding up and then slowing down is verified by the fact that the mean and median value of knickpoint retreat show a trend of increasing and then decreasing as knickpoints retreat upstream; however, the change in the mean and median value of knickpoint retreat with respect to distance is small in BL10 and BL05 relative to BL15. If the authors want to argue this trend occurs in all three experiments, I still believe it would be better to do a statistical test which shows that when fitting a line to the retreat rate data vs distance (for $0<ndd<0.55$ and $0.55<ndd<1$),

the slope of the line is statistically distinct from a slope of 0. I don't think this is a huge deal overall, but it seems like the right thing to do to present a rigorous analysis of the data and to further support the subsequent discussion and interpretations made by the authors.

We had thought during the previous revision that it would sufficient to add the statistics on knickpoint retreat rates in Figure 9. As a remined all the statistical data shown in the Figure 9B were not in the initial ms submission.

In fact, the main reason why the trends are not as clear for the low base level fall experiment (BL05 and potentially BL10; cf Fig 9) as for the high rate one is mainly a matter of scale on the graphs. We initially decided to show all the graphs with the same scale on the y-axis to illustrate the fact that migration rates also depend on base level fall rates. We have now modified again this figure by keeping the same scale on the y-axis only in Fig. 9A but by adjusting it to each data range for each graph in Figures 9B and 9C.

New Figure 9:                                              Former Figure 9:

[Figure]

We also now provide the statistical test required by the reviewer in a new Supplementary Figure (Fig. S4).

[Figure]

I have a number of other minor to moderate comments that I would like the authors to consider:

1) When reading the manuscript I found several English language errors and other word choice that made it difficult to understand what the authors meant. In many other cases there was not sufficient information given about methods or other details, that I think may limit future readers' ability to fully understand the message the authors are trying to convey. I have annotated the PDF with handwritten comments correcting these mistakes and offering other minor comments throughout the text, and attached my comments to this review. I do not expect a line-by-line response to my handwritten comments, but have included them here for the benefit of the authors. I believe these comments will help make a clearer, easier to read manuscript. I think a careful review of the manuscript by the authors to make sure everything is explained in sufficient detail for a reader to understand the analysis and to ensure the wording is clear is necessary before the manuscript can be published.

We sincerely thank the reviewer for his/her efforts to help improve the manuscript. All suggested corrections have been done.

2) I think, but I am not sure, the authors are defining knickpoints two different ways, but only one is listed in the methods. Figure 2 and associated text (L100-105) explains how the authors calculate knickpoints as a single point (the triangles in Fig. 2). However, in subsequent figures, the authors appear to calculate knickpoints as a **zone of discrete length** (e.g., the hatching of K1 and K2 in Fig. 11). It's not clear how this zone is defined. Does this correspond to the circles in Figure 2? Please make this explicit. Similarly, in Figure 7, it's not clear how knickpoint slope is calculated and over what spatial scale this calculation has been made.

We do not understand this comment, hatched areas in Fig 11 do not correspond to lengths but to time! Figure 11 does not show longitudinal profiles but a whole series of parameters calculated along a channel section. We have defined knickpoints in only one way, as described in the methods.

In Figure 7, the slope is calculated from the difference in altitude between the knickpoint lip and knickpoint base, over its extent.